# Loss of HIV candidate vaccine efficacy in male macaques by mucosal nanoparticle immunization rescued by V2-specific response

Systemic vaccination of macaques with V1-deleted (ΔV1) envelope immunogens reduce the risk of $SIV_{mac251}$ acquisition by approximately 60%, with protective roles played by V2-specific ADCC and envelope-specific mucosal IL-17[+]NKp44[+] innate lymphoid cells (ILCs). We investigated whether increased mucosal responses to V2 benefit vaccine efficacy by delivering oral nanoparticles (NPs) that release V2-scaffolded on Typhoid Toxin B (TTB) to the large intestine. Strikingly, mucosal immunization of male macaques abrogated vaccine efficacy with control TTB or empty NPs, but vaccine efficacy of up to 47.6% was preserved with V2-TTB NPs. The deleterious effects of NPs were linked to preferential recruitment of mucosal plasmacytoid dendritic cells (pDCs), reduction of protective mucosal NKp44[+] ILCs, increased non-protective mucosal PMA/Ionomycin-induced IFN-γ[+]NKG2A[-]NKp44[-]ILCs, and increased levels of mucosal activated Ki67[+]CD4[+] T cells, a potential target for virus infection. V2-TTB NP mucosal boosting rescued vaccine efficacy, likely via high avidity V2-specific antibodies mediating ADCC, and higher frequencies of mucosal NKp44[+] ILCs and of ΔV1gp120 binding antibody-secreting B cells in the rectal mucosa. These findings emphasize the central role of systemic immunization and mucosal V2-specific antibodies in the protection afforded by ΔV1 envelope immunogens and encourage careful evaluation of vaccine delivery platforms to avoid inducing immune responses favorable to HIV transmission.

The development of an efficacious vaccine against HIV-1 remains a critical global health objective, but progress has been hindered by a complex array of factors including HIV genetic diversity, gaps in knowledge about HIV host-pathogen interactions, and the challenge of inducing broadly neutralizing antibodies[1]. In 2023, 40 million people were estimated living with HIV, including 1.3 million new infections that year[2]. Although approximately 76% of people living with HIV have access to antiretroviral therapy, the development of an HIV vaccine that is able to minimize the rate of infection for people at risk of acquiring HIV to even 50% could make an important difference in

turning the tide of the current epidemic[3]. To date, around 250 phase I, II, and III clinical HIV vaccine trials have been conducted in humans[4]. Ten trials designed to assess the effectiveness of a vaccine candidate reached phase IIb or III testing[5–15]. Of those, only RV144, a Canarypox-based vaccine, showed statistically significant, albeit moderate, vaccine efficacy (VE)[10], which correlated with the level of antibodies to V1/V2 scaffolded-antigens and V2 peptides[16,17], antibody-dependent cellular cytotoxicity (ADCC)[16], and polyfunctional CD4[+] T cell responses[16]. The modest level of protection observed in RV144 was reproduced in macaques using Canarypox-based SIV vaccines and the primary

e-mail: franchig@mail.nih.gov; berzofsj@mail.nih.gov

correlate of reduced risk was the mucosal level of antibodies to conformational epitopes in V2[18]. More detailed analysis in macaques demonstrated that animals with high levels of V1 antibodies were more susceptible to SIV infection, and deleting V1 from SIV envelope immunogens directly demonstrated that V2 epitopes, but not V1 epitopes, contribute to protection from infection[19]. Extensive work in the macaque model generated a modification of the Canarypox vaccine platform with significantly higher vaccine efficacy[19–21]. Deletion of V1 immunogens, delivered by the DNA/ALVAC/envelope protein/alum vaccine platform (ΔV1DNA/ALVAC/ΔV1gp120/alum), reproducibly decreased the per-challenge risk of SIV$_{mac251}$ acquisition in approximately two thirds of female and male macaques[19–21]. The efficacy of this approach relies on the correct balance of responses with opposite effects on the risk of virus acquisition. Mucosal antibodies to V2[18], specifically antibodies to the coil-helical conformation of V2 mediating ADCC[19–21], CD14[+] cell efferocytosis[20,21], mucosal env-specific IL-17[+] NKp44[+] ILCs[18,21,22], and α4β7[+] CD4[+] T cells negative for CCR5 expression[23] all correlate with decreased acquisition risk. In contrast, antibodies to V1[19], α4β7[+] CD4[+] T cells positive for CCR5 expression[23], and mucosal NKp44[-]NKG2A[-] ILCs producing IFN-γ correlate with increased risk[18,21,22]. These findings improved our understanding of the preclinical immune correlates associated with the ΔV1DNA/ALVAC/ΔV1gp120/alum vaccine regimen and provide a new benchmark to improve vaccine efficacy[20,21].

Here, we hypothesized that the efficacy of systemic immunization with the ΔV1DNA/ALVAC/ΔV1gp120/alum vaccine could be improved by enhanced inoculation of the gut mucosal immune system via oral delivery of a V2 epitope in the appropriate coil-helical conformation. We, therefore, used poly (dl-lactic-co-glycolic acid; PLGA) nanoparticles[24,25] to achieve targeted release of immunogens in the large intestine and rectal mucosa. For V2-TTB NP mucosal immunization, the V2 epitope was scaffolded with typhoid toxin B subunit (TTB)[26] to form a pentamer structure presenting five copies of V2, encapsulated with biologically compatible PLGA nanoparticles[27,28]. This formulation results in the presence of V2 both inside and on the surface of the nanoparticle. To prevent the degradation of nanoparticles in the intestine, NPs were coated with Eudragit FS30D polymer to make 10- to 50-μm microparticles, rendering them too large to be phagocytosed. Coated microparticles release NP at pH 7.4, in contrast to uncoated release at pH 2.5. Thus, upon oral delivery in the microparticle/nanoparticle system, NP uptake occurs almost exclusively in the large intestine. Oral delivery of Eudragit-coated microparticle/nanoparticle formulation has been shown to induce immunity in the colorectal mucosa in mice[24] and in macaques[25].

In the present work, we show that administration of either TTB NPs lacking the V2 epitope or empty NPs abrogate the efficacy of systemic vaccination by recruiting more mucosal plasmacytoid dendritic cells, reducing mucosal NKp44[+] ILCs associated with decreased risk of infection, and by increasing the frequency of mucosal Ki67[+]CD4[+] T cells and IFN-γ[+]NKp44[-]NKG2A[-] ILCs that correlate with increased susceptibility to infection. In animals immunized with V2-TTB NPs carrying V2, the otherwise deleterious effects of the nanoparticle platform inhibiting immune responses and favoring infection were likely counteracted by the increased mucosal and systemic immune responses to V2 (such as increased antibody avidity, V2-specific ADCC, and mucosal B cells secreting IgG and IgA against envelope protein) induced by the V2-TTB NP prime-boost, thereby rescuing vaccine efficacy with a strong trend emerging toward significant protection ($p = 0.053$).

## Results

### Mucosal NP immunization abrogates ΔV1DNA/ALVAC/ΔV1gp120/alum vaccine efficacy

The CKFNMTGLKRDKTKEYNETWYSTDLVCEQGNSTDNESRCYMNHC peptide segment from variable region 2 (V2) of SIV$_{mac251}$ was fused to a structurally compatible joining point on a pentameric scaffold of

typhoid toxin B subunit to form a V2 pentamer (V2-TTB) and formulated with oral nanoparticles (NPs)[24,25]. We immunized thirty male macaques intramuscularly with SIV ΔV1gp160 and p55gag DNAs at weeks 0 and 4, ALVAC-SIV (gag-pro-gp120-TM) at weeks 8 and 12, and boosted with ΔV1gp120 protein formulated in alum at week 12 in the contralateral thigh[19–21]. One group of animals ($n = 12$) was also primed at weeks 0 and 4 and boosted at week 16 with oral V2-TTB NP. A second and third group ($n = 9$ each) were also primed and boosted at the same timepoints with TTB NP or empty NP as controls (Fig. 1a). Eight weeks after the last immunization (week 24), all animals were exposed to up to 11 weekly intrarectal SIV$_{mac251}$ challenges. One additional group of non-immunized animals ($n = 9$) was simultaneously exposed to SIV$_{mac251}$ challenges as naïve control (Fig. 1a). Virus infection was documented by repeated viral load (VL) assays by nanodroplet PCRs of plasma samples. The study design included SIV acquisition data from 18 naïve historical male controls (Materials and methods), data from immunological assays, and acquisition data from 14 animals systemically vaccinated with SIV ΔV1gp160 and p55Gag DNAs, ALVAC-SIV, and ΔV1gp120/alum[19] (Fig. 1a) that were challenged at week 17 with the identical virus stock and in the same animal facility.

No difference in virus acquisition was observed between concurrent and historical controls, as expected ($p = 0.96$; Supplementary Fig. 1a). Systemic vaccination alone decreased the risk of virus acquisition (VE = 57%) compared to the untreated, combined controls, in keeping with prior reports[19] ($p = 0.03$; Fig. 1b). Vaccine+V2-TTB NP demonstrated a strong trend toward decreased risk of virus acquisition compared to the combined controls (VE = 47.6%; $p = 0.053$; Fig. 1c, Supplementary Fig. 1b, c). The addition of oral V2-TTB NP immunization to the regimen elicited no significant improvement in vaccine efficacy compared to the systemic vaccine group ($p = 0.69$, Fig. 1d). Surprisingly, in contrast to the systemic vaccine alone or with V2-TTB NP immunization, a significantly decreased risk of acquisition was not observed in DNA/ALVAC-SIV/ΔV1gp120/alum vaccinated animals prime-boosted with TTB NP without V2 ($p = 0.86$, Fig. 1e, Supplementary Fig. 1d, e) or empty NP ($p = 0.55$, Fig. 1f, Supplementary Fig. 1e, f). Importantly, analysis of peak virus RNA levels in plasma (week 2) and area under the curve for viral load (VL) up to 15 weeks post infection showed that the viral burden in animals that became infected was lower (out to at least 15 weeks) in those vaccinated with V2-TTB NP oral prime-boost compared to naïve controls, as well as to systemically vaccinated animals (Fig. 1g-i).

### Augmented systemic and mucosal antibody responses to V2 by oral V2-TTB NP immunization

We investigated envelope-specific antibody-secreting B cells in the rectal mucosa. B cells secreting both IgG and IgA antibodies binding to ΔV1gp120 were higher in the V2-TTB NP group compared to the TTB NP and empty NP groups at week 19/20 (Fig. 2a, b and Supplementary Fig. 2a, b). Data from the systemically vaccinated group were not available for comparison for this assay.

To further assess the antibody responses to V2, we measured plasma and mucosal IgG binding antibody responses against the most divergent strain, SIV$_{E660}$. We observed significantly higher plasma antibody responses to gp140-SIV$_{E660}$ and a trend toward higher responses to the gp70-SIV$_{E660-BR-CG7V}$-V1/V2 scaffold in the V2-TTB NP group as compared to the TTB NP or empty NP groups (Fig. 2c). Mucosal antibody responses to these proteins were comparable among the groups (Supplementary Fig. 2c). Data from the systemic vaccine group were not available for comparison.

We next measured antibody-mediated ADCC activity in plasma collected prior to the first viral exposure by using ΔV1gp120 protein-coated cells and quantitated the V2-specific ADCC by blockade with mAb NCI05 (Fab')$_2$ that binds the coil-helical conformation of V2[29]. ADCC titers (the reciprocal dilution at which ADCC killing was greater than control killing + 3 SD) were highest in the V2-TTB NP group

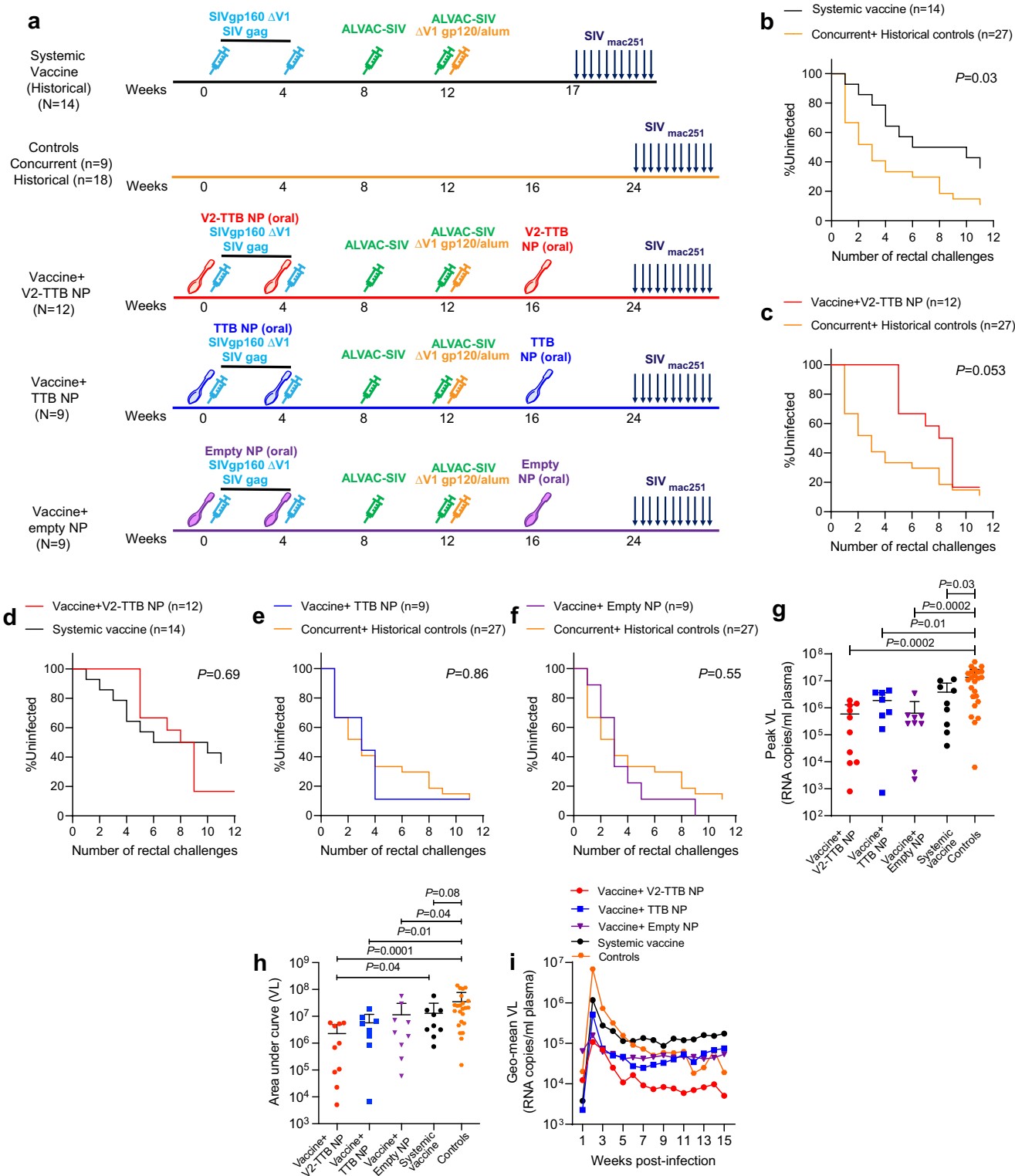

(Fig. 2d), whereas ADCC activity was equivalent in the V2-TTB NP and systemic vaccination groups, and higher than in the TTB NP and empty NP groups (Fig. 2e). Both ADCC titer and ADCC activity correlated with a decreased risk of virus acquisition in the V2-TTB NP group (Fig. 2f, g) as well as in all animals when combined, irrespective of their vaccination regimen (Fig. 2h, Supplementary Fig. 2d).

In order to understand the effect of V2-specific ADCC, we next isolated the F(ab')2 region of a V2 specific monoclonal antibody and used it to block ADCC activity. We used a range of 1 μg to 0.00001 μg serially diluted concentration of F(ab')2 from NCI05, a V2-specific mAb,

and observed that 1 μg of F(ab')2 blocked ADCC more efficiently (Supplementary Fig. 2e). We did not use a higher concentration of F(ab')2 for this assay due to the low yield of F(ab')2 from this antibody. V2-specific ADCC activity was equivalent in the V2-TTB NP and systemic vaccination groups, and higher in both than in the TTB NP and empty NP groups (Fig. 2i). Interestingly, when the frequency of V2-specific ADCC activity was calculated in total ADCC, it was higher in the systemic vaccine group than TTB or empty NP groups (Fig. 2j). To understand the amount of protective V2-specific ADCC activity compared to total or non-V2 specific activity, we compared different ADCC

**Fig. 1 | Schematic representation of immunization regimen, infection rate, and SIV VL. a** Rhesus macaques were subdivided into four groups: vaccine+V2-TTB NP (n = 12), vaccine+TTB NP (n = 9), vaccine+empty NP (n = 9), and concurrent (n = 9) and historical (n = 18) controls. A historical systemic vaccine group (n = 14) was included for immunological comparison. Thirty concurrent animals and fourteen historical animals were primed with DNA-SIV expressing ΔV1gp160 and $SIV_{mac239}$ p55 gag and boosted with ALVAC-SIV encoding *env*, *gag*, and *pol* alone or with ΔV1gp120 protein formulated in alum alhydrogel at the indicated timepoints, and as previously published[19]. Nine animals remained naïve until $SIV_{mac251}$ challenge. At weeks 0, 4, and 16, V2-TTB NP, TTB NP or empty NP were orally administered. For NP administered groups, 5 weeks following the last vaccination (week 24) vaccine efficacy against $SIV_{mac251}$ exposure was assessed by exposing all animals to up to 11 weekly intrarectal viral exposures (arrows) until infection was confirmed. Similarly, for historical systemically vaccinated animals, 5 weeks following the last vaccination (week 17) efficacy against $SIV_{mac251}$ exposure was assessed by exposing all animals to up to 11 weekly intrarectal viral exposures (arrows) until infection was confirmed. **b** Protective vaccine efficacy in the historical systemic vaccine group compared with concurrent+historical controls (p = 0.03). **c** A trend of significant protection in the V2-TTB NP group was observed compared to concurrent+historical controls (p = 0.053). **d** Comparable protection between the V2-TTB NP group and historical systemic vaccine group (p = 0.69). **e, f** No protection was observed in **e** the TTB NP group (p = 0.86) and **f** the empty NP group (p = 0.55) compared to concurrent+historical controls. **g** VL geometric means in different groups (V2-TTB NP, n = 10; TTB NP, n = 8; empty NP, n = 9; systemic vaccine, n = 9; and control, n = 24) of animals 2 weeks post confirmed infection. **h** VL area under curve in different groups (V2-TTB NP, n = 10; TTB NP, n = 8; empty NP, n = 9; systemic vaccine, n = 9; and control, n = 24) of animals up to 15 weeks post confirmed infection. **i** VL geometric means of all macaque groups over time. Data shown in **b**–**f** were analyzed with log-rank (Mantel–Cox) test. Data shown in **g**, **h** were analyzed with two-sided Mann-Whitney test. Horizontal and vertical bars denote mean and SD, respectively. Source data are provided as a Source Data file. Here, the orange, black, red, blue, and purple symbols represent the following groups: male naïve control, male macaques immunized with systemic vaccine, vaccine combined with V2-TTB NP, vaccine combined with TTB NP, and vaccine combined with empty NP, respectively.

activities and observed the lowest amount of V2-specific (NCI05) ADCC compared to total ADCC or non-V2-specific (NCI05) ADCC (Supplementary Fig. 2f), which was expected since V2-specific ADCC was calculated against a single epitope. Furthermore, the ratio of V2-specific ADCC and gp120-specific IgG titer was higher in V2-TTB NP and systemic vaccination groups compared to the empty NP group (Supplementary Fig. 2g). As observed in systemic vaccination[19], V2-specific ADCC correlated with delayed SIV acquisition in the V2-TTB NP group (Fig. 2k) and in all animals combined (Fig. 2l). Furthermore, we observed a negative correlation between V2-specific ADCC and VL at 15 wpi in the systemic vaccine group (Fig. 2m), which further suggested the protective role of this antibody response. However, we did not observe any protective correlation for the non-V2 specific ADCC for the combined animals (Supplementary Fig. 2h). Although V2-specific ADCC activity is lower compared to total and non-V2-specific ADCC, in all cases V2-specific ADCC plays a protective role. Furthermore, TTB-NP or empty NP vaccination decreases total as well as V2-specific ADCC activity, but it is rescued by the presence of V2 responses.

Lastly, we investigated the avidity of the plasma antibody to understand its protective role. The avidity score of antibodies to ΔV1gp120 protein, measured by surface plasmon resonance, trended higher in the V2-TTB NP group than in systemic vaccination (p = 0.056; Fig. 2n). While all four animal groups shared comparable antibody response units, signifying the binding of antibodies to the immobilized antigens, and elevated RU values indicating stronger binding (Supplementary Fig. 2i), they correlated with decreased risk only in the V2-TTB NP group (Fig. 2o). Overall, the data demonstrate that V2-TTB NP oral immunization increased envelope and V2-specific responses both quantitatively and qualitatively in both blood and mucosa.

### Alteration of CD14+ cell activity by oral V2-TTB NP immunization rescues vaccine efficacy

Early work in the SIV model demonstrated that blood myeloid CD14+ cells, which perform efferocytosis, are a reproducible correlate of decreased risk of infection in animals vaccinated with ΔV1 envelope immunogens delivered by the DNA/ALVAC/gp120-alum platform[20,21], and systemic vaccination induces epigenetic change in monocytes that acquire an increased ability to clear apoptotic cells[20]. Since monocytes also engulf nanoparticles[30], we hypothesized that the decreased vaccine efficacy following oral immunization could have affected efferocytosis. We found no difference in the percentage of CD14+ efferocytes among groups (Supplementary Fig. 3a), but efferocytosis MFI, the engulfment of apoptotic cells, was significantly enhanced in the empty NP group compared to V2-TTB NP (MFI of engulfed material in CD14 efferocytes; Supplementary Fig. 3b). Efferocytosis MFI

correlated with decreased risk of infection in the V2-TTB NP group only (Fig. 3a), demonstrating that while V2-TTB oral NP immunization affected the functionality of this monocyte, it ultimately benefitted vaccine efficacy.

We also assessed trogocytosis, in which monocytes nibble surface molecules from adjacent cells and engulf pieces of cell membrane. In the RV306 clinical trial, the ALVAC-HIV vaccine induced high levels of trogocytosis[31], and additional boosts increased the peak magnitude and durability of this immune response[32]. Trogocytosis has been linked to various immune functions, such as immune evasion of target cells, enhanced antigen presentation by effector cells, and the killing of target cells. Deconvolution microscopy of monocytes shows they physically interact with CEM target cells to take up PKH26 without evidence of phagocytosis[33], suggesting our assay is measuring trogocytosis. Interestingly, trogocytosis was lower in all NP-treated groups compared to systemic vaccination only (Fig. 3b). Trogocytosis has been negatively correlated with ADCC in human and macaque studies[34,35]. We observed a positive correlation between trogocytosis and VL at 1 wpi in the V2-TTB NP group (Fig. 3c), suggesting trogocytosis plays a negative role in NP-treated animals. Interestingly, a negative correlation was found between trogocytosis and VL at 8 wpi in the systemic vaccine group (Fig. 3d), suggesting that the protective mechanism underlying efficacy in the systemic vaccine differs from that in NP-treated vaccinated animals.

### Mucosal NPs alter myeloid and plasmacytoid dendritic cell balance in rectal mucosa

Dendritic cells (DCs) are important antigen-presenting cell subsets that coordinate both innate and adaptive immune responses[36]. DCs also efficiently engulf nanoparticles, which might impact their functions, such as maturation, homing, antigen processing, and antigen presentation. Evaluation of myeloid dendritic cells (mDCs) and plasmacytoid dendritic cells (pDCs) in the rectal mucosa (Supplementary Fig. 4) revealed no difference in the percentage of mucosal mDCs at week 17 among the groups (Supplementary Fig. 3c). However, mucosal mDCs negatively correlated with VL at 12 wpi in the V2-TTB group (Fig. 3e), suggesting these cells play a protective role by controlling VL. In contrast, the frequency of mucosal pDCs at week 17 was highest in the TTB NP group, followed by empty NP (Fig. 3f, Supplementary Fig. 3d). Strikingly, pDCs correlated with faster virus acquisition in the TTB NP group (p = 0.03, Fig. 3g) and a positive correlation was observed in the empty NP group with VL at 8 wpi (Fig. 3h). This finding suggests that TTB NPs and empty NPs synergize with pDC engagement. Oddly, the reduced frequency of pDCs in the V2-TTB NP group suggests that the presence of V2 mitigates pDC engagement.

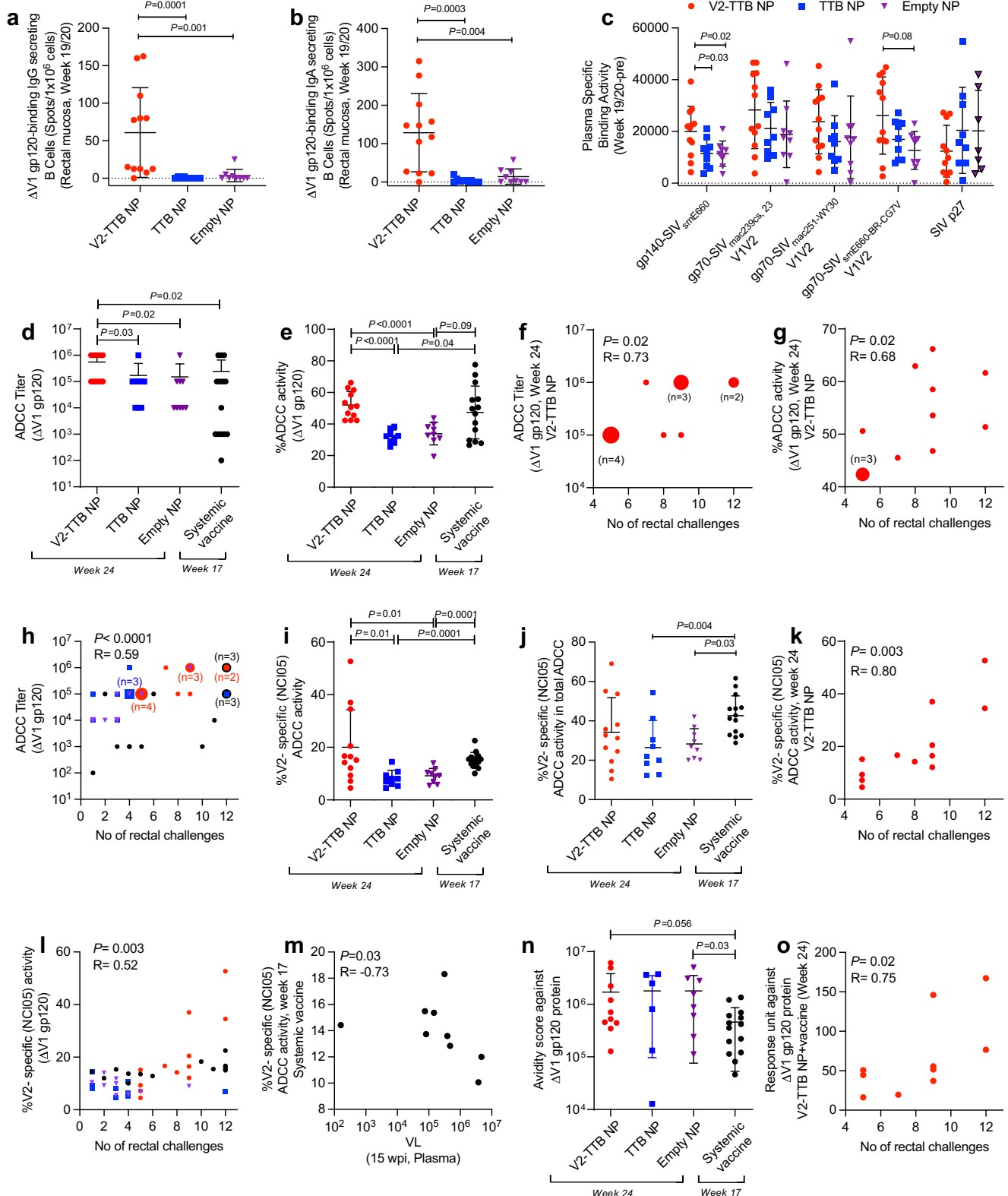

## Mucosal NPs alter the frequency of both NKp44[+] and IFN-γ[+]NKp44[-]NKG2A[-] ILC subsets in rectal mucosa

Systemic vaccination with the ΔV1 immunogens has previously revealed both protective and deleterious roles of different subsets of NK/innate lymphoid cells (ILCs) in SIV acquisition[18,21]. Because nanoparticles are known to enhance NK activity[37], we investigated the frequency of NK/ILCs and their cytokine production in the vaccinated groups (Supplementary Fig. 4). Prior data demonstrated that systemic

vaccination induces mucosal env-specific IL-17[+]NKp44[+] ILCs that correlate with decreased risk of virus acquisition, whereas NKp44[-]NKG2A[-] ILCs producing IFN-γ correlating with increased risk[18,21,22]. In the V2-TTB NP group, the frequency of total NKp44[+] ILCs compared to baseline was increased by vaccination and peaked at week 17 (Supplementary Fig. 3e). At the same timepoint, the prime-boost combination with TTB NPs or empty NPs decreased the frequency of NKp44[+] ILCs compared to the V2-TTB NP group (Fig. 3i and Supplementary

**Fig. 2 | Quantification of humoral responses in plasma and mucosa in rhesus macaques. a, b** Comparison of mucosal B cells in rectal mucosa producing **a** IgG (V2-TTB NP, $n = 12$; TTB NP, $n = 9$; empty NP, $n = 9$) and **b** IgA (V2-TTB NP, $n = 12$; TTB NP, $n = 9$; empty NP, $n = 9$) antibodies that bind to $\Delta$V1 gp120. **c** Comparison of plasma and mucosal IgG binding titer (AUC) against different SIV immunogens in all groups of animals (V2-TTB NP, $n = 12$; TTB NP, $n = 9$; empty NP, $n = 9$). **d, e** Comparison of **d** ADCC titers (the endpoint reciprocal dilution at which the ADCC killing was greater than control killing + 3 SD) and **e** ADCC activity among different groups of animals (V2-TTB NP, $n = 12$; TTB NP, $n = 9$; empty NP, $n = 9$; systemic vaccine, $n = 14$). **f–h** Correlation of **f** ADCC titer in the V2-TTB NP group ($n = 12$, $p = 0.02$), **g** ADCC activity in the V2-TTB NP group ($n = 12$, $p = 0.02$), and **h** ADCC titer in all animal groups combined ($n = 44$, $p < 0.0001$) with number of intrarectal challenges. In **h**, red dots represent V2-TTB NP, black dots represent systemic vaccine, and blue squares represent TTB NP. Enlarged shapes represent multiple animals, as indicated. **i, j** Comparison of **i** V2 (NCI05)-specific ADCC activity and **j** frequency of V2 (NCI05)-specific ADCC activity in total ADCC among different groups of animals (V2-TTB NP, $n = 12$; TTB NP, $n = 9$; empty NP, $n = 9$; systemic vaccine, $n = 14$). **k, l** Correlation of V2 (NCI05)-specific ADCC activity in

the V2-TTB NP group ($n = 12$, $p = 0.003$) and **l** in all animal groups combined ($n = 44$, $p = 0.003$) with number of intrarectal challenges. **m** Correlation of V2 (NCI05)-specific ADCC activity at week 17 with VL at 15 wpi in the systemic vaccine group ($n = 9$, $p = 0.03$). **n** Comparison of plasma antibody avidity score against $\Delta$V1gp120 protein in different groups of animals (V2-TTB NP, $n = 10$; TTB NP, $n = 7$; empty NP, $n = 8$; systemic vaccine, $n = 14$). **o** Correlation of antibody response units against $\Delta$V1gp120 (higher RU values indicated stronger binding of antibody to antigen) with number of intrarectal challenges in the V2-TTB NP+vaccine group ($n = 12$, $p = 0.02$). Data shown in **a–c, e, i, j, n** were analyzed with the two-sided Mann-Whitney test. Data shown in **d** were analyzed with two-sided Cochran-Armitage tests for the comparisons between V2-TTB NP and TTB NP and Empty NP, separately. The two-sided Mann-Whitney test was used for the comparison between V2-TTB NP and systemic vaccine. Data shown in **f–h, k–m, o** were analyzed with the two-sided Spearman correlation test. Horizontal and vertical bars denote mean and SD. Source data are provided as a Source Data file. Here, the black, red, blue, and purple symbols represent male macaques immunized with a systemic vaccine, vaccine combined with V2-TTB NP, vaccine combined with TTB NP, and vaccine combined with empty NP, respectively.

Fig. 3e). Interestingly, the post TTB NP or empty NP immunization frequency of NKp44[+] ILCs decreased at week 17 as compared to week 13, whereas the frequency of these cells was not affected by V2-TTB NP immunization at that timepoint (Supplementary Fig. 3e). Nevertheless, the frequency of NKp44[+] ILCs correlated with decreased risk of virus infection only in the V2-TTB NP group ($p = 0.03$, Fig. 3j). IL-17 plays a role in the formation of tight junctions in the gut epithelium, which in turn help maintain mucosal integrity[38] and might make the gut mucosa less susceptible to viral infection. Since NKp44[+] ILCs are one of the sources of IL-17, the higher frequency of these cells might lead to higher IL-17 expression and protective responses. Taken together, the data suggest that V2-TTB NP immunization impacts mucosal NKp44[+] ILCs differently compared to TTB NP or empty NP immunization.

Since prior work demonstrated that NKp44[-]NKG2A[-] ILCs producing IFN-$\gamma$ correlate with an increased risk of virus acquisition[18,21,22], we also compared their frequency among the different groups. We observed that the oral prime-boost with V2-TTB NP decreased the frequency of NKp44[-]NKG2A[-] producing IFN-$\gamma$, whereas both TTB NP and empty NP increased it (Fig. 3k and Supplementary Fig. 3f). PMA/Ionomycin stimulation induced IFN-$\gamma^+$NKG2A[-]NKp44[-] ILCs positively correlated with VL at 2 wpi in the V2-TTB NP group (Fig. 3l) as well as VL at 4 wpi in the TTB NP group (Fig. 3m). The percentage of IFN-$\gamma^+$NKG2A[-]NKp44[-] ILCs that were induced by PMA/Ionomycin stimulation inversely correlated with myeloid dendritic cell frequency at week 17 in the TTB NP group (Fig. 3n). Overall, these data demonstrate that the oral immunization of systemically vaccinated animals with TTB-NP or empty NP without the V2 scaffolded peptide results in altered balance of mDCs and pDCs as well as NKG2A[-]NKp44[-]IFN-$\gamma^+$ ILCs, likely affecting vaccine efficacy.

## Mucosal NP immunization alters chemokine receptors and homing markers on blood CD4[+] T cells and increases mucosal Ki67[+] CD4[+] T cells

CC-chemokine receptor 5 (CCR5) is required for macrophage-tropic (M-tropic) HIV entry to the cells[39–41] and $\alpha_4\beta_7^{high}$CCR5[+]CD4[+] T cells have been shown to be more susceptible to HIV infection compared to $\alpha4\beta7^{negative}$CCR5[-]CD4[+] T cells[42]. Thus, we investigated the frequency of gut-homing, vaccine-induced (proliferating Ki67[+]) memory CD4[+] T cell subsets expressing or not the CCR5 receptor in the blood of immunized animals collected following the last vaccination.

Vaccine-induced (Ki67[+]) Th2 cells expressing $\alpha_4\beta_7^{high}$CCR5[-] have been associated with decreased risk of SIV acquisition[19–21,23,43]. Thus, we investigated blood $\alpha_4\beta_7^{high}$ CCR5[-] as well as $\alpha_4\beta_7^{negative}$CCR5[-]CD4[+] T cell subsets post-final-vaccination in the systemic vaccination alone and the NP immunized groups. The protective $\alpha_4\beta_7^{high}$CCR5[-] Th2

cells[19–21,23,43] were higher in systemically vaccinated animals compared to the NP-treated animals and were highest in the V2-TTB NP group among all NP immunized animals (Fig. 4a). Importantly, the frequency of $\alpha_4\beta_7^{high}$CCR5[-] Th1 cells was comparable among all NP immunized animals, but it was significantly lower in animals that received systemic vaccination only (Fig. 4b). The frequency of $\alpha_4\beta_7^{high}$CCR5[-] Th17 cells was comparable among all groups (Fig. 4c). The role of the latter two cell subtypes in SIV/HIV infection is not well documented, and thus the immunohistochemistry of tissue re-distribution of these cells requires further study. Taken together, these data suggested that protective $\alpha_4\beta_7^{high}$CCR5[-] Th2 cells were higher in the systemic vaccine group, whereas their level decreases after TTB NP or empty NP vaccination. The incorporation of V2-TTB NPs into the immunization regimen rescued their frequency over the profound suppression by TTB and empty NPs, although the $\alpha_4\beta_7^{high}$CCR5[-] Th2 cells were significantly lower in the V2-TTB NP group than in the systemic immunization group.

Next, we observed that $\alpha_4\beta_7^{negative}$CCR5[-] Th2 cells were higher in both the TTB NP and empty NP groups compared to the V2-TTB NP and systemic vaccine groups (Fig. 4d). Notably, these cells were positively correlated with delayed SIV acquisition only in the TTB group (Fig. 4e). The frequency of $\alpha_4\beta_7^{negative}$CCR5[-] Th1 cells was lowest in the systemically vaccinated group (Fig. 4f), and $\alpha_4\beta_7^{negative}$CCR5[-] Th17 cells were comparable among all groups of animals (Fig. 4g). Since these cells do not express gut homing marker, their implication in mucosal SIV infection is unclear. The frequency of $\alpha_4\beta_7^{high}$CCR5[+] double positive non-protective CD4[+] T cells was very low in all four groups of animals, but the frequency of $\alpha_4\beta_7^{high}$CCR5[+] Th2 (Fig. 4h), $\alpha_4\beta_7^{high}$CCR5[+] Th1 (Fig. 4i), and $\alpha_4\beta_7^{high}$CCR5[+] Th17 cells (Fig. 4j) was higher in the systemic vaccine group than all NP immunized animals.

We next investigated the effect of the various vaccine regimens on vaccine-induced (Ki67[+]) CD4[+] cells in the mucosal tissue. We therefore used immunohistochemistry to assess the frequency of total CD4[+] and Ki67[+]CD4[+] T cells in rectal biopsies collected following the last immunization. Ki67[+]CD4[+] T cells were comparable between V2-TTB and systemic vaccine groups and were found to be significantly lower than in the TTB NP and empty NP group (Fig. 4k, l), demonstrating that NP immunization in the absence of V2 increased the number of activated CD4 T cells, which, in the gut mucosa, are targets for HIV/SIV transmission. These data suggest that the presence of NPs modulated the frequency of systemic, vaccine-induced memory CD4[+] T cells expressing CCR5 and the $\alpha_4\beta_7$ gut-homing marker. However, the impact of NPs on these markers in the mucosa remains uncertain, largely due to the limited availability of biopsies for immune histochemistry.

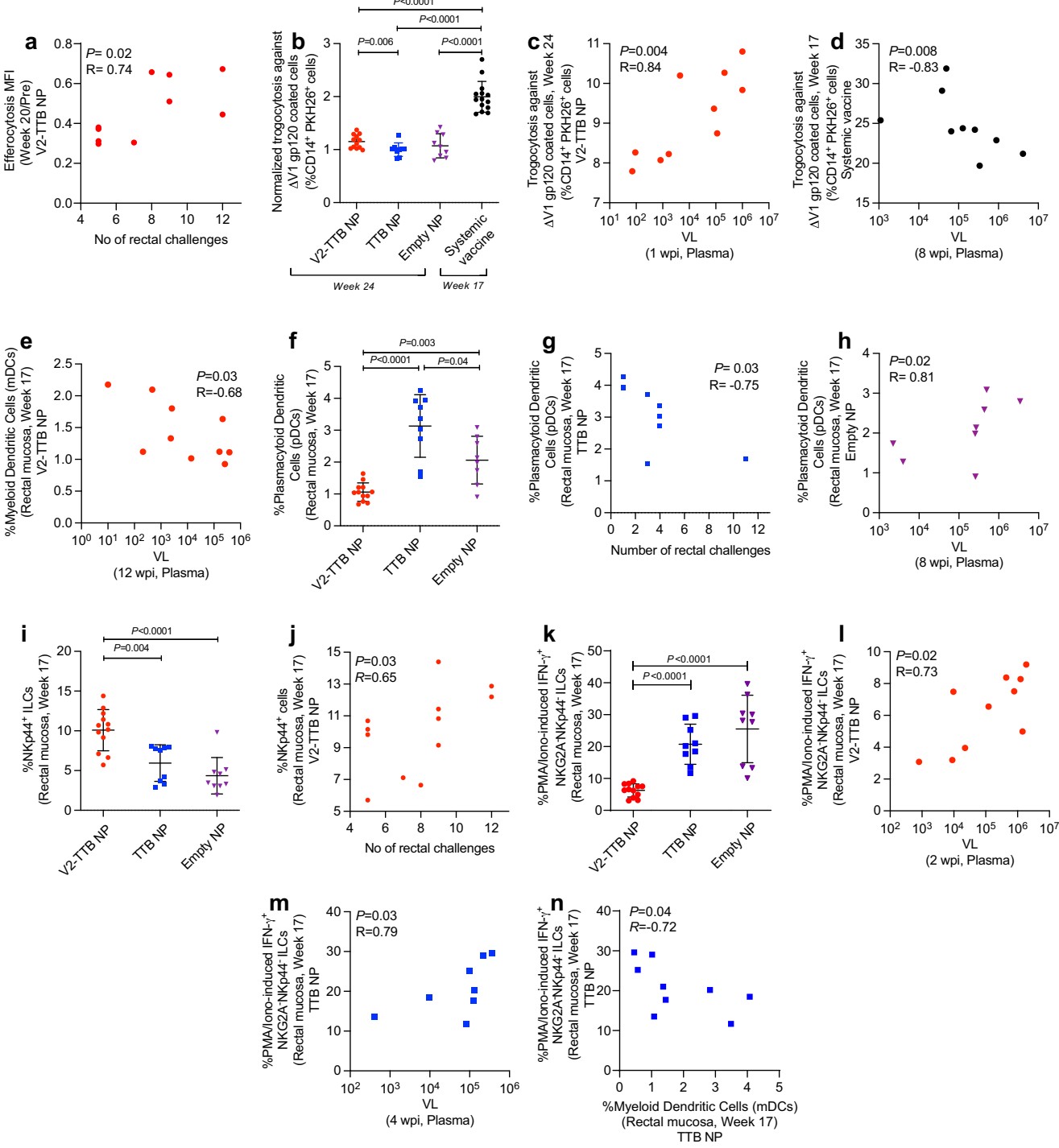

**Fig. 3 | Characterization of systemic cells, mucosal dendritic cells, and mucosal NK cells/ innate lymphoid cells (ILCs) in rhesus macaques. a** Correlation between efferocytosis MFI in blood at week 20/pre and number of intrarectal challenges in the V2-TTB NP group (*n* = 12, *p* = 0.02). **b** Comparison of normalized trogocytosis using antibody-free controls as denominator in plasma against ΔV1 gp120 protein among different groups of animals (V2-TTB NP, *n* = 12; TTB NP, *n* = 9; empty NP, *n* = 9; systemic vaccine, *n* = 14). **c, d** Correlation of trogocytosis in plasma against ΔV1 gp120 protein **c** at week 24 with VL at 1 wpi in V2-TTB NP group (*n* = 10, *p* = 0.004) and **d** at week 17 with VL at 8 wpi in the systemic vaccine group (*n* = 9, *p* = 0.008). **e** Correlation of myeloid dendritic cells in rectal mucosa at week 17 with VL at 12 wpi in V2-TTB NP group (*n* = 10, *p* = 0.03). **f** Comparison of plasmacytoid dendritic cells in rectal mucosa at week 17 in all groups (V2-TTB NP, *n* = 12; TTB NP, *n* = 9; empty NP, *n* = 8). **g, h** Correlation of plasmacytoid dendritic cells in rectal mucosa at week 17 with **g** intrarectal challenges in TTB NP group (*n* = 9, *p* = 0.03) and **h** VL at 8 wpi in empty NP group (*n* = 8, *p* = 0.02). **i** Comparison of NKp44⁺ ILCs

in rectal mucosa at week 17 in all groups (V2-TTB NP, *n* = 12; TTB NP, *n* = 9; empty NP, *n* = 9). **j** Correlation of NKp44⁺ ILCs with number of intrarectal challenges in V2-TTB NP group (*n* = 12, *p* = 0.03). **k** Comparison of PMA/Ionomycin-induced IFN-γ⁺ NKG2A⁻NKp44⁻ ILCs in rectal mucosa at week 17 in all groups (V2-TTB NP, *n* = 12; TTB NP, *n* = 9; empty NP, *n* = 9). **l–n** Correlation of PMA/Ionomycin-induced IFN-γ⁺ NKG2A⁻NKp44⁻ ILCs with **l** VL at 2 wpi in V2-TTB NP group (*n* = 10, *p* = 0.02), **m** VL at 4 wpi in TTB NP group (*n* = 8, *p* = 0.03), and **n** myeloid dendritic cells in rectal mucosa at week 17 in TTB NP group (*n* = 9, *p* = 0.04). Data shown in **b, f, i, k** were analyzed with the two-sided Mann-Whitney test. Data shown in **a, c–e, g, h, j, l–n** were analyzed with the two-sided Spearman correlation test. Horizontal and vertical bars denote mean and SD. Source data are provided as a Source Data file. Here, the black, red, blue, and purple symbols represent male macaques immunized with systemic vaccine, vaccine combined with V2-TTB NP, vaccine combined with TTB NP, and vaccine combined with empty NP, respectively.

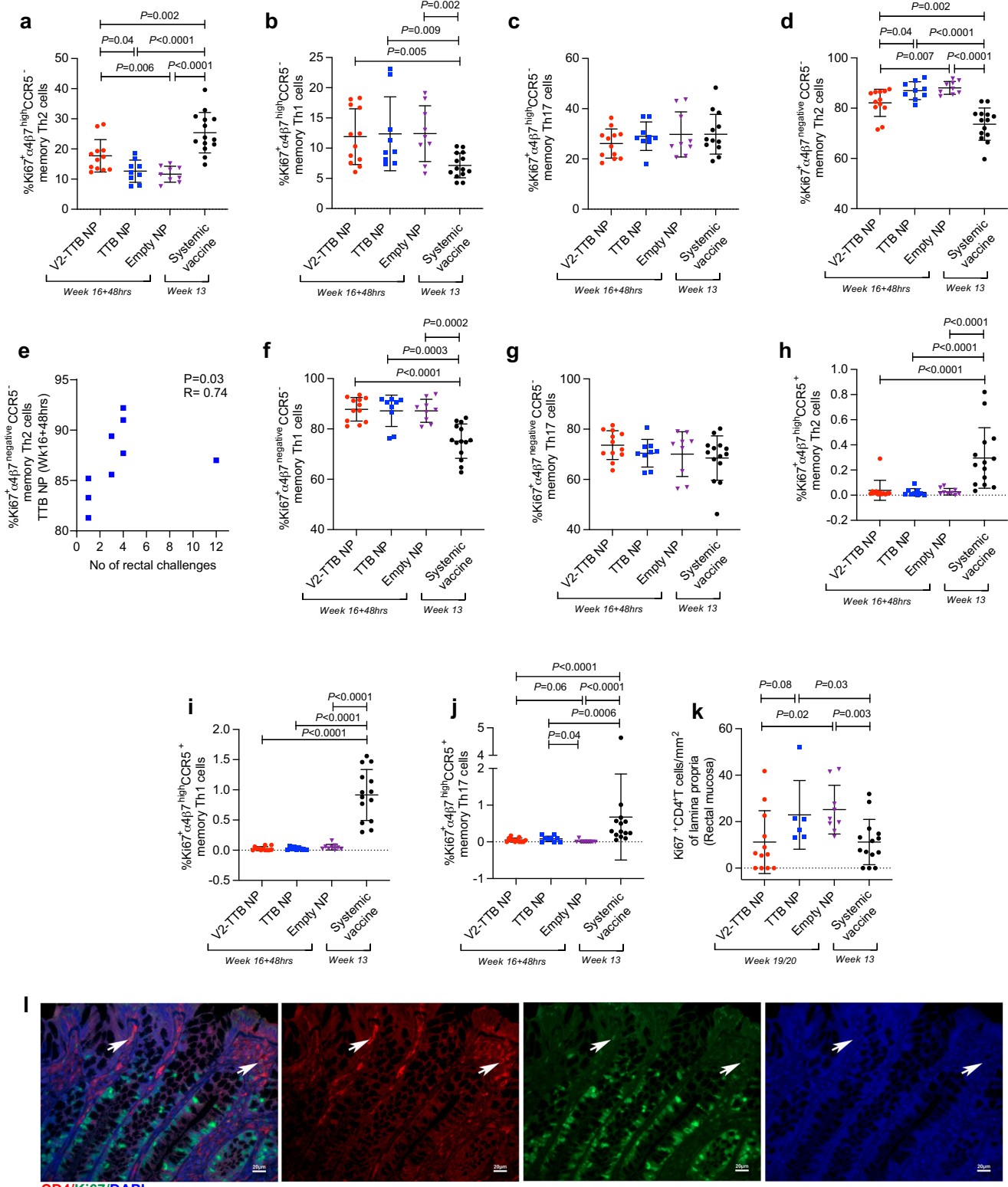

## Mucosal NPs modulate plasma levels of cytokines and chemokines induced by systemic vaccination

Lastly, we evaluated the levels of 45 cytokines/chemokines in plasma collected at baseline and after vaccination from all animal groups. The fold-change of cytokines/chemokines compared to baseline levels was calculated at week 12 + 72 hrs for systemically immunized animals, and at week 16 + 48 hrs for groups vaccinated with NPs. To better visualize the differences of the immune responses induced in the systemically

vaccinated or NP vaccinated animals, principal components analysis (PCA) was performed using as input the fold-change of cytokines/chemokines measured in plasma by proximity assay (Fig. 5a–c). Comparison of the fold-changes between the four groups identified: 1) higher vaccine-induced levels of CXCL-10 and CCL-19 in systemically vaccinated animals compared to all other groups; 2) higher IL-17C and CXCL-11 in systemically vaccinated animals compared to V2-TTB NP and empty NP groups; 3) higher CXCL-12 in systemically vaccinated

**Fig. 4 | Assessment of CD4$^+$ T cells in the blood of rhesus macaques post last vaccination. a–d** Comparison of **a** Ki67$^+$α$_4$β$_7$$^{high}$ CCR5$^-$ memory Th2 cells, **b** Ki67$^+$α$_4$β$_7$$^{high}$ CCR5$^-$ memory Th1 cells, **c** Ki67$^+$α$_4$β$_7$$^{high}$ CCR5$^-$ memory Th17 cells and **d** Ki67$^+$α$_4$β$_7$$^{negative}$ CCR5$^-$ memory Th2 cells among different animal groups (V2-TTB NP, $n = 12$; TTB NP, $n = 9$; empty NP, $n = 9$; systemic vaccine, $n = 14$). **e** Correlation of Ki67$^+$α$_4$β$_7$$^{negative}$ CCR5$^-$ memory Th2 cells at Week16+48hrs with number of intrarectal challenges in TTB NP group ($n = 9$, $p = 0.03$). **f–j** Comparison of **f** Ki67$^+$α$_4$β$_7$$^{negative}$ CCR5$^-$ memory Th1 cells, **g** Ki67$^+$α$_4$β$_7$$^{negative}$ CCR5$^-$ memory Th17 cells, **h** Ki67$^+$α$_4$β$_7$$^{high}$ CCR5$^+$ memory Th2 cells, **i** Ki67$^+$α$_4$β$_7$$^{high}$ CCR5$^+$ memory Th1 cells, and **j** Ki67$^+$α$_4$β$_7$$^{high}$ CCR5$^+$ memory Th17 cells among different groups (V2-TTB NP, $n = 12$; TTB NP, $n = 9$; empty NP, $n = 9$; systemic vaccine, $n = 14$). **k** Comparison of Ki67$^+$ CD4$^+$ T cells per square millimeter of lamina propria in the

mucosa among different groups of animals (V2-TTB NP, $n = 12$; TTB NP, $n = 6$; empty NP, $n = 9$; systemic vaccine, $n = 13$). **l** CD4 (red) and Ki67 (green) double-positive cells with nuclear DAPI (blue) are indicated by arrows in the merged image of CD4, Ki67, and DAPI stains. CD4 (red), Ki67 (green), and DAPI (blue) single stain is also shown (scale bar 20 μm). Data shown in **a–d, f–k** were analyzed with the two-sided Mann-Whitney test. Data shown in (**e**) were analyzed with the two-sided Spearman correlation test. Horizontal and vertical bars denote mean and SD, respectively. Source data are provided as a Source Data file. Here, the black, red, blue, and purple symbols represent male macaques immunized with a systemic vaccine, vaccine combined with V2-TTB NP, vaccine combined with TTB NP, and vaccine combined with empty NP, respectively.

animals compared to the empty NP group; and 4) higher TNFSF-10, CCL-13, TNF, and OSM in systemically vaccinated animals compared to the V2-TTB NP group (Fig. 5a–c, Supplementary Fig. 5). The vaccine-induced changes of all cytokine/chemokines were similar in the TTB NP and empty NP control groups (Supplementary Fig. 5). The majority of differences were observed in the V2-TTB NP or systemic vaccination groups as compared to the remaining three groups (Supplementary Fig. 5).

We then focused on cytokine/chemokine levels measured in plasma collected following the last vaccination (week 16 + 48 hrs) in V2-TTB NP-vaccinated animals to assess whether they correlated with risk of virus acquisition or with other cellular or humoral immune responses. Here, CCL-11 levels negatively correlated with the risk of acquisition in the V2-TTB NP group ($p = 0.03$, Fig. 5d), though overall fold-changes among the four groups were comparable (Supplementary Fig. 6a). Indeed, plasma CCL-11 has been associated with worse HIV outcomes and the loss of CD4$^+$ T cells[44–46]. Interestingly, in the V2-TTB NP group, CCL-11 showed a negative correlation with protective α$_4$β$_7$$^{negative}$CCR5$^-$ memory Th2 cells ($p = 0.03$, Fig. 5e) and V2-specific ADCC activity ($p = 0.04$, Fig. 5f), further suggesting CCL-11 is associated with SIV acquisition.

Since CXCL-10 plays a role in the recruitment of plasmacytoid dendritic cells to inflamed tissue[47], and in this study the mucosal pDC frequency was significantly lower in the V2-TTB NP group compared to TTB NP and empty NP groups (Fig. 3f), we investigated whether the levels of CXCL-10 measured following vaccination in the V2-TTB NP group were associated with dendritic cell frequencies. Interestingly, we identified a significant negative correlation between mucosal pDC and CXCL-10 in the V2-TTB NP group ($p = 0.02$, Fig. 5g). Thus, we conclude that the level of plasma CXCL-10 might play a role in the lower level of mucosal pDC seen in the V2-TTB NP group, which might in turn contribute to protection in the V2-TTB NP group.

Additionally, Lymphotoxin-alpha (LTα) is a member of the tumor necrosis factor (TNF) superfamily and might be a contributor to HIV pathogenesis[48]. Post vaccination fold-changes compared to baseline among the four groups were comparable for LTα (Supplementary Fig. 6b). Here, LTα levels in the V2-TTB NP group measured at week 16 + 48 hrs were negatively associated with the number of challenges required to infection ($p = 0.045$, Fig. 5h). Activated mucosal CD4$^+$ T cells might act as a target for HIV/SIV infection[18,49,50], thus we expected a positive correlation between mucosal activated CD4$^+$ T cells and LTα. However, we observed a negative correlation between mucosal-activated CD4$^+$ T cells and LTα ($p = 0.02$, Fig. 5i). While the LTα level might contribute to susceptibility to acquisition, it does not appear to be through induction of mucosal-activated CD4$^+$ T cells as targets of infection.

IL-15 is one of the most important cytokines for regulating innate as well as adaptive immune responses[51,52], though the relevant mechanics of this cytokine in HIV infection are debated. Some reports suggest that IL-15 is important to control HIV/SIV/SHIV infection[53,54] whereas others have suggested that IL-15 plays a pathogenic role[55,56]. In the current study, we observed a comparable fold-change of IL-15

among animal groups (Supplementary Fig. 6c) and the level of Il-15 post last vaccination was not associated with SIV acquisition. However, in the V2-TTB NP group IL-15 showed a positive correlation with non-protective PMA/Ionomycin-induced IFN-γ$^+$ NKG2A$^-$NKp44$^-$ ILCs ($p = 0.02$, Fig. 5j), suggestive of an indirect role for IL-15 in viral acquisition in the current HIV/SIV macaque model.

IL-7 augments HIV-1 replication in vitro[57,58] and in HIV infected individuals IL-7 levels has been associated with increased viral load[59]. Here, we observed a negative correlation between IL-7 and V2-specific ADCC activity ($p = 0.04$, Fig. 5k), further suggesting the non-protective role of IL-7 in SIV acquisition.

## Discussion

ΔV1DNA/ALVAC/ΔV1gp120/alum vaccination in both male and female macaques has exhibited a range of 57-70% vaccine efficacy along with statistically significant protection from viral acquisition[19–21]. Based on the hypothesis generated by multiple studies in humans and non-human primates that V2-specific antibodies play an important role in protection from HIV and SIV[16,19–21,60,61], and that most transmission in humans is through mucosal routes (genital and gastrointestinal), it follows that induction of V2-specific antibodies in the genital and gastrointestinal mucosa should be an important goal of HIV vaccines. Since immunization with even a single SIV linear B-cell epitope was shown to generate protective immune responses in macaques[43], incorporating V2 peptides into a mucosal immunization regimen is a logical approach to achieve mucosal immunity. To achieve this goal, TTB-scaffolded V2-pentamers in NPs were designed to augment the levels of V2-specific antibodies and antigen-specific antibody-producing B cells in the mucosa. Indeed, in our study the V2-TTB NP group had the highest level of ΔV1gp120-binding B cells in rectal pinch biopsies (Fig. 2 a, b) as well as the highest avidity score against ΔV1gp120 (Fig. 2n) and highest ADCC titer against ΔV1gp120 (Fig. 2d). ADCC was performed using same human donor PBMCs as effector cells, gp120 coated using EGFP-CEM-NKr-CCR5-SNAP cells as target cells, and frozen plasma as source of antibody. Thus, the differences observed in the study are not attributed to the PBMCs or target cells. Furthermore, PBMCs from different donors may affect the frequency of ADCC killing. Due to donor-to-donor variability, the ADCC killing rate would proportionally adjust based on the specific PBMCs used. Thus, data from different donors should be normalized for accurate comparison. To account for this variability, testing multiple donors with a sufficient number of plasma samples is necessary. However, using PBMCs from multiple donors is aimed at reducing inter-experiment variability, which can also be controlled by using a single donor with a larger sample size. In this study, we used PBMCs from a single donor for all ADCC assays, ensuring that the data presented are consistent and comparable.

Nanoparticles up to 200 nm in size can be engulfed by dendritic cells[62] and can present antigens effectively, but they could also generate a tolerogenic DC phenotype which impacts adaptive and innate immunity[63]. Moreover, the size of the NP can impact DC functionality. Our goal was to augment mucosal immunity by delivering

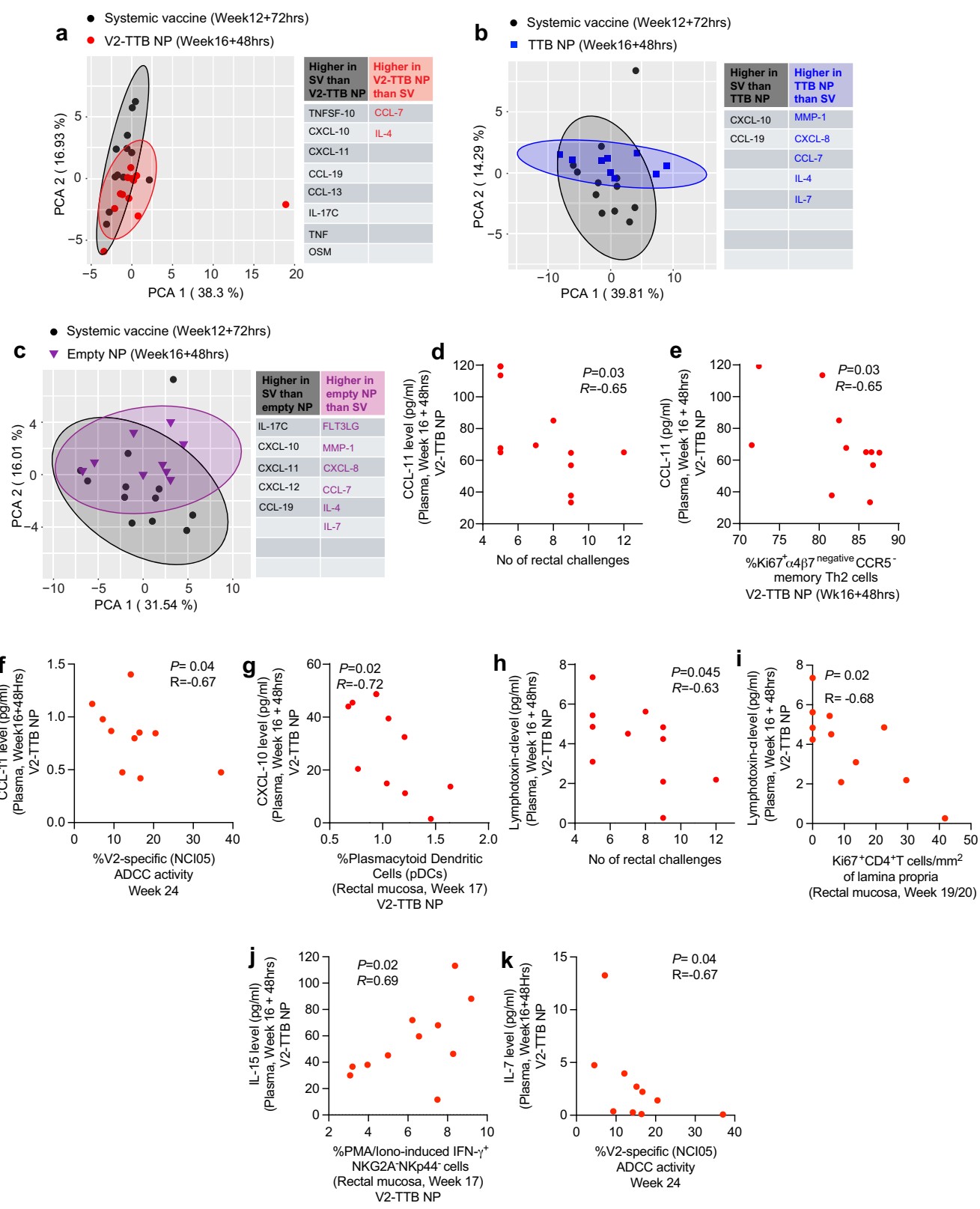

nanoparticles containing TTB scaffolded V2 peptides to the large intestine to protect against intrarectal SIV challenge. Surprisingly, NPs without the V2 antigen were found to have a negative effect on vaccine efficacy (Fig. 1e, f). TTB-NPs and empty NPs preferentially recruited mucosal pDCs (Fig. 3f), whose level correlated with faster virus acquisition (Fig. 3g) and a reduction in the frequency of protective NKp44[+] ILCs (Fig. 3i). Furthermore, NPs increased non-protective

responses such as mucosal NKp44[-]NKG2A[-] ILCs producing IFN-γ and mucosal-activated Ki67[+]CD4[+] T cells. Systemic vaccination with the ΔV1DNA/ALVAC/ΔV1gp120/alum vaccine regimen alone decreases the risk of virus infection in approximately two-thirds of vaccinated animals[19]. However, in the current study systemic vaccination combined with mucosal immunization with V2 antigen scaffolded to NP (V2-TTB NP) showed a vaccine efficacy of 47.6%, and the combination

**Fig. 5 | Evaluation of plasma cytokine correlation with different immune parameters in V2-TTB NP group. a–c** Principal component analysis comparing fold-change of 45 cytokines/chemokines in the plasma of NP-treated vaccinated animals at 48 hrs after last immunization (week 16 + 48 hrs) and systemically vaccinated animals at 72 hrs after last immunization (week 12 + 72 hrs) compared to baseline. Comparisons of systemic vaccine and **a** V2-TTB NP, **b** TTB NP, and **c** empty NP. **d–f** Correlation of CCL-11 level at week 16 + 48 hrs with **d** number of intrarectal challenges ($n = 11$, $p = 0.03$), **e** Ki67$^+$ α$_4$β$_7$$^{negative}$ CCR5$^-$ memory Th2 cells ($n = 11$, $p = 0.03$) and **f** V2-specific ADCC activity ($n = 11$, $p = 0.04$). **g** Correlation of CXCL-10 level at week 16 + 48 hrs with mucosal plasmacytoid dendritic cells at week 17 ($n = 11$, $p = 0.02$). **h, i** Correlation of Lymphotoxin-alpha (LTα) level at week 16 + 48 hrs with

**h** number of intrarectal challenges ($n = 11$, $p = 0.045$) and **i** Ki67$^+$ CD4$^+$ T cells per square millimeter of lamina propria in the mucosa ($n = 11$, $p = 0.02$). **j** Correlation of IL-15 level at week 16 + 48 hrs with mucosal PMA/Ionomycin-specific IFN-γ$^+$ NKG2A$^-$NKp44$^-$ ILCs at week 17 ($n = 11$, $p = 0.02$). **k** Correlation of IL-7 level at week 16 + 48 hrs with V2-specific ADCC activity at week 24 ($n = 11$, $p = 0.04$). Principal component analysis **a–c** was conducted using R studio to explore data variability. Data shown in **d–k** were analyzed with the two-sided Spearman correlation test. Source data are provided as a Source Data file. Here, the black, red, blue, and purple symbols represent male macaques immunized with systemic vaccine, vaccine combined with V2-TTB NP, vaccine combined with TTB NP, and vaccine combined with empty NP, respectively.

did not decrease the infection risk compared to systemic vaccine alone ($p = 0.69$; Fig. 1d). Further, once infected after multiple intrarectal challenges, the animals that received V2-TTB NPs had the lowest peak viral load as well as the lowest VL area under the curve 15 weeks after infection of any group, including the systemically vaccinated animals (Fig. 1g–i). The increased virus acquisition following administration of TTB-NPs and empty NPs is reminiscent of the concerns raised from the beginning of the AIDS epidemic that a vaccine might be a double-edged sword, inducing activated CD4$^+$ T cells at the site of infection that might induce protective immunity but also serve as target cells for virus[18,49,50]. Inducing such responses at the site of entry of the virus, namely the mucosal challenge site, might have even sharper effects in both dimensions.

Mucosal innate immune responses, such as DCs, NK/ILCs, and monocyte/macrophages, can influence the outcome of HIV/SIV infection. Myeloid and plasmacytoid dendritic cells are two of the major DC subsets[64]; mDCs recognize diverse pathogens by their broad TLR expression, while pDCs mainly recognize +ssRNA and unmethylated CpG DNA motifs by TLR7 and TLR9, respectively[36]. In vitro studies showed that mDC from people living with HIV-1 are functionally defective[65,66] and that bystander activation of pDCs[67] might contribute to mDC dysfunction, suggesting that the balance between pDCs and mDCs is important in the control of HIV/SIV infection. Similarly, the balance of mucosal NKp44$^+$ ILCs and mucosal NKG2A$^-$NKp44$^-$ ILCs appears to be key in protection from SIV infection and disease[18,21,22,68]. Systemic CD14$^+$ classical monocytes are also important in protecting against SIV infection[20,23,69], but their role has not been explored. To investigate the mechanisms responsible for altered vaccine efficacy observed when combining systemic vaccination with empty NPs, TTB NPs, or V2-TTB NPs, we closely analyzed the mucosal immune responses. Compared to the V2-TTB NP group, the frequencies of mucosal pDCs (Fig. 3f), mucosal IFN-γ$^+$ NKG2A$^-$NKp44$^-$ ILC (Fig. 3k), and mucosal Ki-67$^+$ CD4$^+$ T cells (Fig. 4k) were increased in the TTB NP and empty NP groups, but not the V2-TTB NP group. Notably, all of these responses have been associated with increased risk of SIV/HIV acquisition[18,68,70–72]. On the other hand, mucosal NKp44$^+$ ILCs[18,21,22,68] and blood CD14$^+$ cells are correlated with decreased risk of virus infection in both male and female macaques[20,23,69]. The frequency of mucosal NKp44$^+$ ILCs was also lower in the TTB NP and empty NP groups compared to V2-TTB NP group (Fig. 3i). Taken together, these findings indicate that mucosal immunization with NPs lacking V2 antigen increases non-protective responses while decreasing protective ones (Fig. 6). However, the presence of the V2 immunogen on the NP was nonetheless able to mitigate the adverse responses generated by mucosal immunization with empty NPs and TTB NPs. One possible explanation for this finding is that the increased level and avidity of V2 antibodies and ADCC may have compensated for the deleterious immune responses by opsonizing NPs and/or modifying innate responses. The avidity of the V2 antibody may play a critical role, especially in view of SIV's low envelope spike density, which reduces the benefit of antibody bivalency and makes ADCC activity more dependent on the intrinsic affinity. More specifically, V2 might have increased protective NKp44$^+$ ILCs and decreased the Ki67$^+$ CD4$^+$ T cells

that could serve as targets for virus entry (Fig. 6). In addition, V2-TTB NPs increased the local mucosal antigen-specific IgG and IgA-secreting B cells in the rectal mucosa (Fig. 2a, b), likely contributing to protection.

V2-specific ADCC activity was associated with VL in systemic vaccination (Fig. 2m), but no such correlation was observed for NP treated groups. Furthermore, trogocytosis was positively associated with VL in the V2-TTB NP group (Fig. 3c), and negatively associated with VL in the systemic vaccine (Fig. 3d), suggesting this antibody-dependent monocyte response acts differently in viremia control and/or promotion in different vaccine regimens. Furthermore, mDC frequency was associated with VL control in the V2-TTB NP group (Fig. 3e), whereas pDC promoted increased VL in the TTB NP group (Fig. 3h). Interestingly, PMA/Ionomycin-induced IFN-γ$^+$ NKG2A$^-$NKp44$^-$ ILCs promoted increased VL in the V2-TTB and TTB groups (Fig. 3l, m). Taken together, these data suggested that different vaccine regimens promote distinct immune responses, which might influence the VL once animals become infected.

The CCR5 receptor on CD4 T cells is essential for macrophage-tropic (M-tropic) HIV entry into cells[39–41]. One of the immune correlates of reduced risk elicited by the DNA/ALVAC/gp120/alum vaccine platform is Th2 cells expressing no or low levels of CCR5[19–21,23,43]. We observed here that "protective" Ki67$^+$ α$_4$β$_7$$^{high}$ CCR5$^-$ memory Th2 cells were higher in the systemically vaccinated group (Fig. 4a). However, the frequency was lower in TTB NP and empty NP groups but elevated in the V2-TTB NP group (Fig. 4a), suggesting the additional V2 immunization also affected the CD4$^+$ T cell phenotype. In contrast, Ki67$^+$α$_4$β$_7$$^{negative}$ CCR5$^-$ Th2 cells were highest with TTB NP and empty NP immunization (Fig. 4d) and these cells correlated with delayed SIV acquisition in the TTB NP group (Fig. 4e). Thus, NP immunization also changed the balance of α4β7 and CCR5 markers on memory CD4$^+$ T cells in the blood of the vaccinated animals and influenced the outcome of vaccination.

All together, these data reinforce that systemic and mucosal antibodies to V2 levels are key correlates of protection from HIV/SIV infection and suggest that boosting mucosal antibodies may increase the efficacy of an HIV vaccine. On the other hand, they emphasize the complexity of immune responses to HIV/SIV and highlight how platforms used to deliver HIV immunogens must be tested individually and experimentally in suitable animal models to minimize the risk of unexpected immune modulation favoring HIV transmission. Ultimately our use of an NP carrying an immunogen, in this case one that directs the immune response to V2, shows that it is possible to both enhance mucosal responses and target them to a particular antigen. The findings made in this study offer tangible data that could advance development of an efficient mucosal vaccine that not only inhibits mucosal transmission of HIV-1, but also overcomes the platform's inherent deleterious effects.

## Methods

The research complies with all relevant ethical regulations. The NCI Animal Care and Use Committee (ACUC) approved the vaccine study.

**Fig. 6 | Nanoparticle modulation of mucosal immune responses.** The administration of V2-TTB NPs (red circle), TTB NPs (blue circle), and Empty NPs (purple circle) differently affect mucosal immune responses. Administration of TTB NPs (center) or Empty NPs (right) decreases protective responses such as V2-specific antibody responses, antigen-specific B cell responses, and frequency of NKp44⁺ ILCs. In contrast, non-protective immune responses such as plasmacytoid dendritic cells (pDCs) and IFN-γ⁺ NKG2A⁻NKp44⁻ ILCs (IFN-γ⁺ double negative [DN] cells) increase in those animal groups. In the V2-TTB NP group (left), these protective responses were higher, whereas non-protective responses were lower. This figure was created by F. Bhuyan with BioRender.com. (2023) BioRender.com/n41m055.

## Animals

Thirty-nine male Indian rhesus macaques obtained from the free-range breeding colony on Morgan Island, South Carolina, were used in this study. The macaques, aged 3 to 4 years at study initiation, were negative for SIV, simian retrovirus, and STLV. The macaques were randomized by the statistician based on age, weight and before being divided into 4 groups: vaccine+V2-TTB NP ($n = 12$); vaccine+TTB NP ($n = 9$); vaccine+Empty NP ($n = 9$); and control ($n = 9$). These group sizes are based on a previous vaccine study comparing 14 vaccinated macaques and 18 controls[19].

In this study, 14 historical systemically vaccinated animals were included for comparison[19]. These macaques were immunized in the same manner as the rest of the current NP-vaccinated groups, but they did not receive any NP immunization. The animals were maintained in the same facility as the animals in the current study and challenged with the same SIV viral stock as the current vaccinated groups. Furthermore, data on 18 historical naïve controls intrarectally challenged with the same viral stock of $SIV_{mac251}$ and at the same dilution ($TCID_{50}$) were added to the control group to increase statistical power. There was no difference between the concurrent and historical controls in the survival rate of SIV acquisition ($p = 0.96$).

If the animals are given up to 11 viral challenges and the numbers of challenges to infection are compared using the score test of the proportional hazards model (equivalent to log rank test), then at the two-sided $p \leq 0.05$ level with infection rate of 0.108 between the vaccine group of 12 or 9 animals and the control group of 27 animals, the power of the test is expected to be 35.3% (vaccine efficacy 44.8%) and 27.4% (vaccine efficacy 43.7%), respectively. While with an infection rate of 0.065 between the vaccine group of 12 or 9 animals and the control group of 27 animals, the power of the test is expected to be

82.5% (vaccine efficacy 66.9%) and 71.5% (vaccine efficacy 66.4%), respectively. With an assumption of lower infection rate for the combination of vaccine and nanoparticles, a group size of 12 or 9 animals were added to different arms of the vaccine groups. The power calculations and the averaged vaccine efficacies (VE) are based on 10,000 simulated datasets where the infection rates in the historical control and the historical vaccine are 0.21 and 0.09, respectively.

The aim of the study was to assess vaccine efficacy in macaques immunized with systemic vaccination and the mucosal immunization combination. A further goal was to compare the immune responses and vaccine efficacy to prior experiments involving male macaques receiving systemic vaccines and naïve male controls. For this purpose, male macaques were selected for the current study.

Macaques were housed and maintained at the NCI Animal Facility at the National Institutes of Health, Bethesda, MD. All animals were handled in accordance with the standards of the Association for the Assessment and Accreditation of Laboratory Animal Care (AAALAC) in an AAALAC-accredited facility (OLAW, Animal Welfare Assurance A4149-01 for NIH). All animal care and procedures were carried out under protocols approved by the NCI Animal Care and Use Committee (ACUC) prior to study initiation. Animals were closely monitored daily for any signs of illness, and appropriate medical care was provided as needed. Animals were socially housed per the approved ACUC protocol and social compatibility except during the viral challenge phase when they were individually housed. All clinical procedures, including biopsy collection, administration of anesthetics and analgesics, and euthanasia, were carried out under the direction of a laboratory animal veterinarian. Steps were taken to ensure the welfare of the animals and minimize the discomfort of all animals used in this study. Animals were fed daily with a fresh diet of primate biscuits, fruit, peanuts, and other

food items to maintain body weight or normal growth. Animals were monitored for psychological well-being and provided with physical enrichment including sanitized toys, destructible enrichment (cardboard and other paper products), and audio and visual stimulation.

## Nanoparticle preparation

The V2 loop-TTB pentamer and TTB scaffold were designed and provided by Susan Zolla-Pazner and Xiang-Peng Kong, Mt. Sinai and NYU, respectively. V2loop(M766)-TTB was codon optimized for mammalian cell expression and synthesized by a commercial vendor (Genscript, NJ) in the modified expression vector pVRC8400 (kindly provided by the Vaccine Research Center, National Institutes of Health) with a secretion signal and C-terminal 8X His tag. Plasmid DNA was then transiently transfected with polyethyleneimine (PEI) into FreeStyle 293-F cells (Thermo Fisher Scientific), and cell supernatants containing the secreted protein were purified with nitrilotriacetic acid (Ni-NTA) agarose. The nanoparticle formulation had previously been adapted for oral administration in macaques[25] and all nanoparticles were produced by James Talton, Alchem Laboratories Corporation. The nanoparticles were reconstituted at 90-100 mg powder/dose in 5 mL diluted lime juice (1:30). Each dose contained approximately 300 µg of V2 loop-TTB pentamer, TTB scaffold or empty NP for the V2-TTB NP, TTB NP or empty NP groups, respectively. The nanoparticle suspensions were shaken at Room Temperature (RT) for at least 1 hour then delivered by feeding tube into the stomach, followed by a small volume of air and a 10 mL saline rinse.

## Immunization and challenge

Macaques in the V2-TTB NP+vaccine group (12 macaques), TTB NP +vaccine group (9 macaques) and Empty NP+vaccine group (9 macaques) groups were immunized at weeks 0 and 4 with DNA encoding $SIV_{M766}$ ΔV1gp160 (2 mg/dose) and $SIV_{mac239}$ gag (1 mg/dose) in a total volume of 1 ml PBS. DNA was administered in both thighs (0.5 ml to each). At 8 weeks the macaques were administered ALVAC encoding *gag/pro/env* (wildtype *env*) in the right thigh, $10^8$ pfu/dose in 1 ml PBS. At week 12 the macaques were boosted with the same ALVAC plus $SIV_{M766}$ ΔV1gp120 protein (400 µg/dose in 500 µl PBS plus 500 µl 2% Alhydrogel). The ALVAC was administered to the right thigh; the 1 ml dose of Env protein plus alum was administered to the left thigh. With the help of a feeding tube, the animals of these three groups were orally administered with V2-TTB NP, TTB NP or empty NP at week 0, 4 and 16. Beginning at week 24 all macaques were weekly challenged intrarectally with 1 ml of a $SIV_{mac251}$ (stock day 8 from 2010) $TCID_{50}$ 400 (calculated in Rhesus 221 cells). Up to 11 challenges were administered until the macaques became SIV positive as assessed by droplet digital PCR (H.K. Chung, J. Narola, H. Babbar, M. Naseri, N. Richardson, R. Pal, and T. Fouts, manuscript in preparation). Rhesus macaques were vaccinated, and samples were collected during the vaccination period as well as during and after challenging the animals with SIV $_{mac251}$.

## IgG plasma titers to gp120

gp120 total IgG antibodies were measured by ELISA. ELISA plates (Nunc Maxisorp 96 well plate) were coated with 100 µl of 500 ng/ml $SIV_{mac251-M766}$ gp120 protein /well in 50 mM sodium bicarbonate buffer pH 9.6 and incubated overnight at 4 °C. Plates were blocked with 200 µl PBS Superblock (Thermo Fisher Scientific, Waltham, Massachusetts, USA) for 1 h at room temperature (RT). Plasma samples were serial-diluted with sample diluent (Avioq, Durham, North Carolina, USA), and 100 µl of diluted plasma was added to the wells. Plates were covered and incubated for 1 h at 37 °C, washed 6 times with PBS Tween 20 (0.05%), and incubated with 100 µl of anti-human HRP diluted at 1:120,000 in sample diluent (Avioq) for 1 h covered at 37 °C. The plates were washed 6 times. Plates were developed using 100µl K-Blue Aqueous substrate (Neogen, Lexington, Kentucky, USA) to all wells and incubated for 30 min at RT. The reaction was stopped by the addition

of 100 µl of 2 N Sulfuric acid to all wells and the plate was read at 450 nm on a Molecular Devices (San Jose, California, USA) E-max plate reader.

## Binding antibody multiplex assay (BAMA)

Week 19/20 plasma and rectal mucosal IgG binding antibody responses were evaluated against SIV proteins by binding antibody multiplex assay (BAMA)[16,60,73,74]. A panel of five SIV proteins (gp140-$SIV_{smE660}$, gp70-$SIV_{mac239cs,23}$ V1V2, gp70-$SIV_{mac251WY30}$ V1V2, gp70-$SIV_{SME660-BR-CG7V}$ V1V2, SIV p27) were conjugated to MagPlex® magnetic beads (Luminex Corporation, Austin Texas) and mixed with 1:80 dilution of plasma samples, titrated 5-fold for 6 dilutions, assayed in duplicate. Rectal sponges were processed[75], with resulting rectal secretions tested at 1:2 and 1:20 dilutions. IgG binding was detected using goat anti-monkey IgG-biotin (Rockland Immunochemicals, Inc., Pottstown, Pennsylvania, USA) followed by PE Streptavidin (BD Pharmingen, San Diego, California, USA). Beads were analyzed on a Bio-Plex 200 instrument, with binding magnitude expressed as mean fluorescence intensity (MFI). IgG purified from a $SIV_{mac251}$-infected rhesus macaque was included as an assay positive control and blank (uncoupled), and empty gp70 scaffold (MuLV gp70) conjugated beads as negative controls. For plasma samples, a response was considered positive if the MFI was (1) greater than 100, (2) greater than the antigen-specific cutoff (95th percentile of all pre-immunization sample binding to the antigen), and (3) 3-fold higher than the matched pre-immunization sample before and after blank bead or empty gp70 scaffold bead subtraction. Plasma binding antibody titer is reported as area under the titration curve, calculated over the dilution series (1:50, 1:250, 1:1250, 1:6250, 1:31250, 1:156250) for MFI for each sample per antigen, using the trapezoidal method. For mucosal samples, SIV-specific BAMA binding MFI values were normalized to total IgG, as quantified by the Meso Scale Discovery® (MSD) Human/NHP Isotyping Panel 1 Kit (Meso Scale Diagnostics LLC, Rockville, Maryland, USA), and computed as specific activity (SA) (SA = BAMA MFI*dilution divided by total IgG in µg/mL). Mucosal samples were considered positive if the sample (1) MFI was greater than 100, (2) SA greater than the antigen-specific cutoff (95th percentile of all pre-immunization sample SA per antigen), and (3) SA greater than 3-fold that of the matched pre-immunization sample.

## ADCC CEM-based assay

ADCC activity was assessed[19–21,68,69] using EGFP-CEM-NKr-CCR5-SNAP cells that constitutively express GFP as targets[76]. Briefly, one million target cells were incubated with 50 µg of ΔV1gp120 protein for 2 h at 37 °C. After this coating, the target cells were washed and labeled with SNAP-Surface® Alexa Fluor® 647 (New England Biolabs, Ipswich, Massachusetts, USA) per manufacturer recommendations for 30 min at RT. Previously frozen plasma samples, heat-inactivated at 56 °C for 30 min, were serially diluted (7 ten-fold dilutions starting at 1:10) and 100 µl were added to wells of a 96-well V-bottom plate (Millipore Sigma, St. Louis, Missouri, USA). 5000 target cells (50 µl) and 250,000 human PBMCs (50 µl) were added as effectors to each well to give an effector/target (E/T) ratio of 50:1. The plate was incubated at 37 °C for 2 h followed by two PBS washes. The cells were resuspended in 200 µl of a 1% PFA solution and acquired on a Symphony equipped with a high throughput system (BD Biosciences). Specific ADCC activity was measured by loss of GFP from the SNAPAlexa647+ target cells. Target and effector cells cultured in the presence of R10 medium were used as background. Note that only the plasma antibodies were from each individual animal; the effector and target cells were constant for all the assays. Anti-SIVmac gp120 monoclonal antibody KK17 (NIH AIDS reagent program) was used as a positive control. Normalized ADCC activity was calculated as: (ADCC activity in the presence of plasma – background)/(ADCC activity in the presence of KK17 – background) × 100. The normalization was done to minimize plate to plate and experiment to experiment variation of the assay. The ADCC endpoint

titer is defined as the reciprocal dilution at which the percent ADCC activity was greater than the mean percent ADCC activity of the background wells containing medium only with target and effector cells, plus three standard deviations[77–79].

## Inhibition of ADCC CEM-based assay by monoclonal F(ab')2 of NCI05

F(ab')2 fragments were prepared from both NCI05 mAb, as these antibodies recognize overlapping conformationally distinct V2 epitopes[19], using Pierce f(ab')2 Micro Preparation Kit (cat. #44688, Thermo Fisher) following the manufacturer's instructions. An SDS-page gel with the recovered F(ab')2 was run and Silver stained (cat. #LC6070, Silver Quest staining Kit, Invitrogen, Waltham, Massachusetts, USA) according to the manufacturer's instructions, to assure the purity of the F(ab')2 fragments. V2-specific ADCC activity was assessed as follows[19–21]. Target cells, coated with ΔV1gp120 protein as indicated above and labeled with SNAP-Surface® Alexa Fluor® 647, were incubated for 1 h at 37 °C with 5 μg/ml of purified F(ab')2 fragments from NCI05 monoclonal antibody. Cells incubated without F(ab')2 served as control. These target cells were subsequently used in the ADCC assay as described above. These F(ab')2 inhibit binding (and ADCC) mediated by the anti-V2 antibodies from immunized animals' plasma. The percentage ADCC activity difference in the presence or absence of F(ab')2 is considered V2 specific ADCC activity.

## Surface plasmon resonance (Biacore)

Antibody avidity determinations were conducted using the Biacore 4000 surface plasmon resonance (SPR) system[43,80,81]. Briefly, the immobilizations were performed using a standard amine-coupling kit. The CM-5 sensor chip surface was activated with a 1:1 mixture of 0.4 M 1-ethyl-3-(3-dimethylaminopropyl) carbodiimide hydrochloride (EDC, Cytiva) and 0.1 M N-hydroxysuccinimide (NHS, Cytiva) for 600 sec. Protein $SIV_{M766}$ ΔV1gp120 (20 μg/mL each) in 10 mM sodium acetate pH 4.5 was immobilized to spots 1, 2, 4, and 5 of flow cell 1 on the CM5 sensor chip (Cytiva). The resulting Response Units (RUs) were as follows: 13734-14197 RU. Spot 3 was left unmodified to serve as a reference. Following the surface preparation, heat-inactivated (56 °C for 45 minutes) plasma samples were diluted 1:50 in running buffer (10 mM HEPES, 150 mM NaCl, 0.05% Tween-20, pH7.4) and injected onto the protein immobilized surface for 250–350 sec followed by dissociation for 1300-2400 sec. Data for each sample were collected at a rate of 10 Hz, with an analysis temperature of 25 °C. All sample injections were conducted at a flow rate of 10 μL/min. The bound surface was regenerated with 150 mM HCl for 60 seconds. Data analysis was performed using Biacore 4000 Evaluation software 4.1 with double subtractions for the unmodified surface and buffer blank. Fitting was conducted using the dissociation mode integrated with Evaluation software 4.1 (GE Healthcare, Uppsala Sweden). The data are shown as RU, and avidity score. Avidity score was calculated as RU/kd.

## Trogocytosis

Trogocytosis was measured using CEM cell line[34,43]. CEM.NKR.CCR5 cells were washed with PBS and stained with PKH26 (Sigma-Aldrich, St-Louis, MO, USA) at 2 μM in Diluent C at RT for 5 min. Cells were then washed with R-10, resuspended in R-10, and incubated with ΔV1 gp120 for 1 hr at RT in 96-well polypropylene plates. The assay was performed with an antibody-free control. Cells were washed twice with R-10 and incubated with 300-fold diluted plasma samples. Cryopreserved healthy control PBMC were next added in R-10 at an effector to target (E:T) cell ratio of 50:1 and then incubated for 5 h at 37 °C. After the incubation, cells were washed, stained with live/dead aqua fixable stain and anti-CD14 APC-H7 (clone MΦP9, BD, San Jose, CA, USA), washed again, and fixed with 4% PFA (Tousimis, Rockville, MD). Fluorescence was evaluated on an LSRII flow cytometer (BD Biosciences). Trogocytosis was evaluated by measuring the %PKH26⁺ from the live CD14⁺

cells. Results were normalized using the no antibody control for comparison between different animal groups.

## ELISpot

Lymphocytes from rectal pinch biopsies were quantified by ELISpot for ΔV1gp120, specific IgG and IgA secreting B cells[82] at pre-vaccination, 3 weeks post fourth vaccination (week 15) and 3/4 weeks post last vaccination (week 19/20). Freshly collected rectal biopsies were digested with collagenase (2 mg/ml; Sigma-Aldrich) in the absence of FBS in 37 °C for 1 hour, then it was mechanically separated by using a 10 ml syringe with a blunt head canula. It was washed with R10 and passed through 70 μm cell strainer. Single cells were counted and used for the experiment[21,68]. Briefly, 96-well multi-Screen-IP Filter Plates (0.45 μm; Millipore Sigma) were activated with 70% ethanol, washed three times with PBS, and coated overnight, for at least 18 hours prior to the experiment. Plates were stored at 4 °C until use. To quantify Env-specific IgG and IgA antibody-secreting cells (ASC), wells were coated in 300 ng/well of ΔV1gp120. Lymphocytes from rectal mucosa were added ($2 \times 10^5$/well in duplicate) and incubated at 37 °C overnight. After washing with PBS containing 0.05% Tween-20 (Sigma) (PBST), plates were blocked with 200 μl/well R-10 plus 3% BSA (Sigma) for 2 hr at 37 °C. After washing with PBST, the plates were incubated with 1 μg/ml of biotinylated anti-mouse IgG or IgA (Rockland) in PBST + 1% FBS for 2 hr at 37 °C. Plates were then washed four times with PBST and incubated with HRP-Avidin conjugate (Vector Laboratories) in PBST + 1% FBS for 1 hr at RT. To develop, the plates were first washed three times with PBST, three times with PBS, and then 3 amino-9 ethyl-carbazole (AEC, Sigma) was added for approximately 3 minutes. Plates were washed twice with water and allowed to dry overnight before imaging with the ImmunoCapture Program. All spots were manually counted and the mean number of spots from duplicate wells were recorded.

## Immunohistochemistry stain of Ki67⁺CD4⁺ T cells in rectal mucosa

Immunohistochemistry was performed using frozen rectal mucosa tissue fixed in 4% PFA and embedded in paraffin. Rectal mucosa from 41 animals was collected at week 13 for systemically vaccinated animals and at week 19/20 for nanoparticle-vaccinated animals. A total of 41 rectal mucosa biopsy was analyzed. The primary antibodies were mouse (IgG2a) monoclonal anti-CD4 (clone OTI5D9, Novus Biologicals) and rabbit anti-Ki67 (Abcam). Normal rabbit IgG and mouse IgG (Invitrogen) were included with staining serving as the negative control. The binding of the primary antibodies was detected simultaneously using Goat anti-Mouse IgG2a Cross-Adsorbed Secondary Antibody, Alexa Fluor™ 568, Invitrogen™, and Goat anti-Rabbit IgG (H + L) Highly Cross-Adsorbed Secondary Antibody, Alexa Fluor™ Plus 488, Invitrogen™ (Fisher Scientific). Nuclei were stained with 4',6-diamidino-2-phenylindole dihydrochloride hydrate (DAPI). Digital images were captured and analyzed using the Zeiss Axiocam System and Openlab software (Inprovision)[83,84]. Multiple layer images with CD4, Ki67, and DAPI stains in separate layers were captured on 5 randomly selected fields covering 472,100 μm² of tissue. One transparency reference layer was generated to mark CD4, Ki67, and CD4/Ki67 double-positive cells with DAPI nuclear stain. A merged image of CD4, Ki67, and DAPI stain was used to eliminate unspecific stains. The following numbers of animals were used for their respective groups: V2-TTB NP ($n = 12$), TTB NP ($n = 6$), empty NP ($n = 9$), and systemic vaccine ($n = 14$). The numbers of positive cells are presented as cells per square millimeter of lamina propria.

## Efferocytosis assay

The frequency of efferocytotic CD14⁺ cells was assessed by Efferocytosis Assay kit (#601770, Cayman Chemical company, Ann Arbor, MI, USA)[20,21,43]. CD14⁺ cells were used as effector cells, whereas

apoptotic neutrophils were used as target cells. The protocol was readapted in order to use CD14[+] monocyte cells rather than differentiated macrophages due to the low cell availability. CD14[+] cells were isolated from cryopreserved PBMCs ($10 \times 10^6$ cells) collected following Pre-immunization and 3-4 weeks post last immunization (week 19/20) by using non-human primate CD14 MicroBeads (#130-091-097, Miltenyi Biotec Inc.) and following manufacturer instructions. At the end of the separation, cells were counted and stained with CytoTell Blue™ provided in the kit and following manufacturer instructions. One unrelated macaque was used as source of neutrophils as target cells. Neutrophils were isolated[85] as follows, after isolation of PBMCs by Ficoll Plaque (GE Healthcare), the bottom red pellet was added to an equal volume of 20% dextran in water, gently mixed, and incubated for 1 min. Approximately three volumes of PBS were added, mixed again, and incubated in the dark for 50-60 minutes. At the end of incubation, the clear layer at the top of the tube containing neutrophils was collected. Cells were pelleted and treated with ACK lysing buffer (Quality Biological, Gaithersburg, MD, USA) for 5 min at 37 °C, washed with R10 and counted. Neutrophils were stained with CFSE provided in the kit and following manufacturer instructions. Apoptosis of neutrophils was induced by treatment with Staurosporine Apoptosis inducer provided in the kit. Briefly, isolated cells were resuspended in R10 containing Staurosporine diluted 1:1000 and incubated at 37 °C for 3 hours. At the end of the incubation cells were washed twice with R10 and used for the efferocytosis assay. Subsequently, effector and apoptotic target cells were cultured alone (as controls) or cocultured at a ratio of one effector CD14[+] cell and three target apoptotic neutrophils. Cells were incubated at 37 °C for 12 hours. At the end of the coculture, cells were washed with PBS, fixed with 1% PFA in PBS, and acquired on a FACSymphony A5 and examined using FACSDiva software (BD Biosciences) by acquiring all stained cells. Data were further analyzed using FlowJo v10.1 (BD Biosciences). The frequency of efferocytotic CD14[+] cells was determined as the frequency of double-positive cells for CytoTell™ Blue and CFSE on the CytoTell™ Blue positive monocytes.

## Rectal mucosal NK/ILC, monocyte and dendritic cell phenotyping

The frequency of NK/ILC and dendritic cells were measured in macaque rectal mucosa pre vaccination, 1 week post fourth vaccination (week 13), and 1 week post last vaccination (week 17). A 6 mm biopsy forceps was used to obtain 15 pinch biopsies from the rectal mucosa, and 11 of the freshly collected rectal biopsies were digested with collagenase (2 mg/ml; Sigma-Aldrich) in the absence of FBS in 37 °C for 1 hour, then mechanically separated by using a 10 ml syringe with a blunt head canula. It was washed with R10 and passed through 70 µm cell strainer. Single cells were counted and used for the experiment[21,68]. From 10 to 15 million cells were recovered from the pinches. Two million cells were used for phenotype. For flow cytometry analysis, this is the only proven/known method to isolate cells from rectal mucosa. We have tried to isolate cells by mechanical process only, absent enzymatic process, but this approach fails to produce high-quality single cells. A similar observation was reported by Bondonese et al.[86]. The data for all animals are comparable since all animals went through the same process of tissue collection and the same methods were used to isolate the cells. Cells were stained with Live/Dead blue dye (cat. #L34962, 0.5 µl) from Thermo Fisher, followed by surface staining with the following: Alexa 700 anti-CD3 (SP34-2; cat. #557917, 5 µl), Alexa 700 anti-CD20 (2H7; cat. #560631, 5 µl), BV510 anti-CD11c (3.9; cat. #748269, 5 µl), BV650 anti-NKp44 (P44-8; cat. #744302, 5 µl), BV786 anti-CD45 (D058-1283; cat. #563861, 5 µl), BUV395 anti-CD123 (7G3; cat. #564195, 5 µl), BUV496 anti-CD16 (3G8; cat. #612944, 5 µl), BUV661 anti-HLA-DR (G46-6; cat. #612980, 5 µl), BUV805 anti-CD14 (M5E2; cat. #565779, 5 µl), from BD Biosciences (San Jose, California, USA); PE-Cy7 anti-NKG2A (Z199; cat. no. B10246, 5 µl) from Beckman Coulter and APC-Cy7 anti-CD11b (ICRF44; cat. #47-0118-42, 5 µl) from Thermo

Fisher (Waltham, MA, USA) for 30 minutes at room temperature. Samples were acquired on a BD FACSymphony A5 cytometer and analyzed with FlowJo software 10.6. Myeloid dendritic cells were gated as singlets, live cells, CD45[+] cells, CD14[-], HLA-DR[+,] CD11c[+] (Supplementary Fig. 3). Plasmacytoid dendritic cells were gated as singlets, live cells, CD45[+] cells, CD14[-], HLA-DR[+], CD123[+] (Supplementary Fig. 3). NKG2A[+] NK cells were gated as singlets, live cells, CD45[+] cells, CD3[-], CD20[-], CD11b[-], and NKG2A[+] NKp44[-] cells. NKp44[+] cells were gated as singlets, live cells, CD45[+] cells, CD3[-], CD20[-], CD11b[-], and NKG2A[-] NKp44[+] cells. NKG2A[-] NKp44[-] cells were gated as singlets, live cells, CD45[+] cells, CD3[-], CD20[-], CD11b[-], and NKG2A[-] NKp44[-] cells (Supplementary Fig. 3).

## Rectal mucosal NK/ILC cytokine expression upon gp120 peptides/PMA stimulation in vaccinated animals

The cytokine levels of NK/ILCs were measured in macaque rectal mucosa pre vaccination 1 week post fourth vaccination (week 13) and 1 week post last vaccination (week 17). A 6 mm biopsy forceps was used to obtain 15 pinch biopsies from the rectal mucosa and 11 of the freshly collected rectal biopsies were digested with collagenase (2 mg/ml; Sigma-Aldrich) in the absence of FBS in 37 °C for 1 hour, then it was mechanically separated by using a 10 ml syringe with a blunt head canula. It was washed with R10 and passed through a 70 µm cell strainer. Single cells were counted and used for the experiment. The two million cells were cultured in R10 in the presence/or absence of gp120 peptides or PMA/Ionomycin for 18 hours. Subsequently, cells were stained with Live/Dead blue dye (cat. #L34962, 0.5 µl) from Thermo Fisher, followed by surface staining with the following: Alexa 700 anti-CD3 (SP34-2; cat. #557917, 5 µl), Alexa 700 anti-CD20 (2H7; cat. #560631, 5 µl), BV650 anti-NKp44 (P44-8; cat. #744302, 5 µl), BV786 anti-CD45 (D058-1283; cat. #563861, 5 µl), from BD Biosciences (San Jose, California, USA); PE-Cy7 anti-NKG2A (Z199; cat. no. B10246, 5 µl) from Beckman Coulter and APC-Cy7 anti-CD11b (ICRF44; cat. #47-0118-42, 5 µl) from Thermo Fisher (Waltham, MA, USA) for 30 minutes at RT. This was followed by permeabilization with a FOXP3-transcription buffer set (cat. #00-5523-00) from eBioscience (San Diego, California, USA) according to the manufacturer's recommendation and subsequently intracellular staining with the following: BV421 anti-IFN-γ (B27; cat. #562988, 5 µl) from BD Biosciences and PE-Cy5.5 anti-IL-17 (BL168; cat. # 512314, 5 µl) from BioLegend (San Diego, California, USA) for 30 minutes at RT. Samples were acquired on a BD FACSymphony A5 cytometer and analyzed with FlowJo software 10.6. Cytokines were gated on parent population.

## CD4[+] T cell phenotypes

The frequency of CD4[+] T cell subsets was measured in blood at baseline and week 13 in vaccinated animals. PBMCs were stained with the following: LIVE/DEAD™ Fixable Blue Dead Cell Stain (cat. #L23105, Thermo Fisher); Alexa 700 anti-CD3 (SP34-2; cat. #557917, 5 µl), BV785 anti-CD4 (L200; cat. #563914, 5 µl), PeCy5 anti-CD95 (DX2; cat. #559773, 5 µl), BV650 anti-CCR5 (3A9; cat. #564999, 5 µl), BUV496 anti-CD8 (RPA-T8; cat. #564804, 5 µl), and FITC anti-Ki67 (B56; cat. #556026, 5 µl) from BD Biosciences; APC Cy7 anti-CXCR3 (G025H7; cat. #353722, 5 µl), and BV605 anti-CCR6 (G034E3; cat. #353420, 5 µl), from BioLegend; and APC anti-α4β7, provided by the NIH Nonhuman Primate Reagent Resource (R24 OD010976, and NIAID contract HHSN272201300031C). Samples were acquired on a BD FACSymphony A5 cytometer and analyzed with FlowJo software 10.6. Gating was done on live CD3[+]CD4[+] cells and on vaccine-induced Ki67[+] cells. Memory cells were identified as CD95[+] cells. CXCR3 and CCR6 expression were used to identify Th1 or Th2 or Th17 CD4[+] T cell populations[23].

## Proximity Extension Assay (PEA) on plasma samples

Protein quantification was executed employing the Olink® Target 48 Cytokine panel* (Olink Proteomics AB, Uppsala, Sweden) in

accordance with the manufacturer's protocols. This method leverages the Proximity Extension Assay (PEA) technology, as extensively detailed by *Assarsson et al.*[87]. This specific PEA methodology enables the concurrent assessment of 45 distinct analytes. Briefly, we used pairs of oligonucleotide-labeled antibody probes, each tailored to selectively bind to their designated protein targets. Probe pairs mix were incubated with 1 µl of plasma. Probes that encountered their cognate proteins are then in close spatial proximity and their respective oligonucleotides engage in pair-wise hybridization. A DNA polymerase was used to amplify the polymerized DNA, and to create distinct PCR target sequences. Subsequently we detected and quantified these newly formed DNA sequences through utilization of a microfluidic real-time PCR platform, specifically the Biomark HD system by Fluidigm (Olink Signature Q100 instrument). Data validation to uphold data integrity was conducted with the Olink NPX Signature software specifically designed for the Olink® analysis: the application was used to import data from the Olink Signature Q100 instrument and process the data. Data normalization procedures were executed employing an internal extension control and calibrators, thereby effectively mitigating any inherent intra-run variability. The ultimate assay output was reported in picograms per milliliter (pg/ml), predicated upon a robust 4-parameter logistic (4-Pl) fit model, thereby ensuring precise absolute quantification. Comprehensive insights into the assay's validation parameters, encompassing limits of detection, intra- and inter-assay precision data, and related metrics are available at www.olink.com. Principal component analysis was conducted using R studio to explore data variability and detect underlying patterns. Fold-changes between baseline and subsequent time points were calculated for each biomarker. For the PCA, numeric foldchanges biomarker data were scaled to ensure equal variance contribution. FactoMineR package was used for the PCA to calculate principal components and determining the variance explained by each. The first two principal components were extracted providing the basis for data visualization. ggplot2 was used for graphical representation, displaying the PCA results.

## Statistical analysis

Statistical analysis was performed without testing normal distribution and equal variances of the data and therefore non-parametric tests were used. Two-tailed Wilcoxon signed-rank test or two-tailed Mann-Whitney test was used to compare continuous factors between two paired or unpaired groups, respectively. Comparisons of differences between groups in the number of challenges before viral acquisition were assessed using the log-rank (Mantel-Cox) test of the discrete-time proportional hazards model. The average per-risk challenge of viral acquisition was estimated as the total number of observed infections divided by the number of administered challenges. Correlation analyzes were performed using the non-parametric Spearman-rank correlation method. Since this research was conducted as exploratory, all p-values are reported as nominal values without adjusting for multiple comparisons. Initially no animals or data points were excluded from the analyzes. However, based on the reviewers' request, any mucosal cell of interest with fewer than 300 events were excluded from the analysis.

## Reporting summary

Further information on research design is available in the Nature Portfolio Reporting Summary linked to this article.

## Data availability

Source data are provided as a Source Data file. Source data are provided with this paper.

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

## Acknowledgements

We thank D. Ahern for editorial support. We thank Dr. F. Bhuyan for creating the summary figure-6. We gratefully acknowledge J. Kramer, M. Breed, W. Magnanelli, M. Metrinko, K. Killoran, and their staff for expert care of the rhesus macaques and collection of all tissues at the NCI animal facility. We thank D. Venzon for randomization of the animal groups. We thank Sabrina Helmold Hait for training in the antigen-specific B cell, plasma cell and plasmablast assays. We thank K. McKinnon and S. Brown (Vaccine Branch, NCI) for flow cytometry support. We thank N.R. Kanchetty for providing support in performing efferocytosis assay. We thank N. Eisel, T. Peters, A. Odeh, and L. Zhang for BAMA data acquisition and/or analysis. We thank S. Shen for her contribution in plasma antibody response analysis. The following reagent was obtained through the NIH AIDS Reagent Program, Division of AIDS, NIAID, NIH: APC anti-$\alpha_4\beta_7$ (A4B7R1; cat. #051514AB) by the NIH Nonhuman Primate Reagent Resource (R24 OD010976, and NIAID contract HHSN272201300031C). This work was supported with federal funds from the National Cancer Institute Intramural Research program (to G.F. and J.A.B.) and from the Office of AIDS Research (#ZIA-BC-004020 to G.F. and J.A.B.), National Institutes of Health. This work was supported in part by a cooperative agreement (W81XWH-18-2-0040) between the Henry M. Jackson Foundation for the Advancement of Military Medicine, Inc., and the U.S. Department of Defense (DOD), as well as NIH HHSN272201100016 (G.T.), 5P30-AI064518 (G.T.), DP1 DA036478 (T.C.) and R01 AI145655 (X.-P.K and S.Z.-P.) as well as support from the Department of Medicine, Icahn School of Medicine at Mount Sinai, New York (S.Z.-P.).

## Author contributions

The study was conceived by G.F., J.A.B., T.C., S.Z.-P., Y.S., and M.A.R.; M.A.R. and G.F. wrote the first draft of the manuscript; T.C., J.A.B., S.Z.-P., and all co-authors contributed to writing the final draft of the manuscript; M.A.R. prepared the vaccine and challenge virus, processed samples, and coordinated and conducted the macaque studies alongside M.B.; M.A.R. conducted the ADCC assay, mucosal innate response analysis, phenotyping, and intracellular cytokine assays, maintained collaborations, analyzed the data, and prepared the figures; M.B. conducted efferocytosis and CD4$^+$ T cell analysis, and contributed to data analysis; H.S. obtained the animals and their grouping with balanced age and weight, collected the V2-TTB from collaborators and arranged its incorporation into nanoparticles, coordinated macaque studies, and processed tissue and blood specimens; H.S. and S.E.H. performed mucosal antigen-binding B cell responses by ELISpot assays; S.S. and A.G. conducted mucosal innate response experiments and sample processing; Z.M. performed immunohistochemistry; X.J., C.C.L., and X.P.K. cloned and produced the V2-TTB and TTB constructs; L.S. conducted O-Link assays and performed PCA analysis; I.S.C. conducted the historical systemic vaccine study and generated F(ab')2 fragments for ADCC assay; S.B. and M.R. performed Surface Plasmon Resonance (Biacore) assays; K.F.N. and D.P.-P. conducted trogocytosis assays; M.B.-F. and T.C. performed antibody titer assays against V2 peptides; L.D.W. and G.D.T. designed and conducted BAMA assays; R.M., M.D., T.H., E.W., and Y.S. processed and preserved samples and contributed to the immunological assays; H.C.-W. conducted confirmatory statistical analysis; T.C. and S.Z.-P. contributed to the design of different NP constructs; S.Z.-P. assayed antibodies against various V1-V2, V2, and gp120 constructs; J.D.T. generated NPs.

## Funding

## Competing interests
The authors declare no competing interests.

## Additional information

Mohammad Arif Rahman [1,13], Massimiliano Bissa [1,13], Hanna Scinto [2,13], Savannah E. Howe[2], Sarkis Sarkis[1], Zhong-Min Ma [3], Anna Gutowska [1], Xunqing Jiang[4], Christina C. Luo [4], Luca Schifanella[1], Ramona Moles[1], Isabela Silva de Castro[1], Shraddha Basu[5,6], Kombo F. N'guessan[5,6], LaTonya D. Williams[7,8], Manuel Becerra-Flores[9], Melvin N. Doster[1], Tanya Hoang[2], Hyoyoung Choo-Wosoba[10], Emmanuel Woode[1], Yongjun Sui [2], Georgia D. Tomaras[7,8], Dominic Paquin-Proulx [5,6], Mangala Rao [5], James D. Talton[11], Xiang-Peng Kong[4], Susan Zolla-Pazner [12], Timothy Cardozo [9], Genoveffa Franchini [1,14] ✉ & Jay A. Berzofsky [2,14] ✉

[1]Animal Models and Retroviral Vaccines Section, Vaccine Branch, Center for Cancer Research, National Cancer Institute, National Institutes of Health, Bethesda, MD, USA. [2]Vaccine Branch, Center for Cancer Research, National Cancer Institute, National Institutes of Health, Bethesda, MD, USA. [3]California National Primate Research Center, University of California, Davis, Davis, USA. [4]Department of Biochemistry and Molecular Pharmacology, NYU Grossman School of Medicine, New York, NY, USA. [5]United States Military HIV Research Program, CIDR, Walter Reed Army Institute of Research, Silver Spring, MD, USA. [6]Henry M. Jackson Foundation for the Advancement of Military Medicine, Inc., Bethesda, MD, USA. [7]Center for Human Systems Immunology, Department of Surgery, Duke University School of Medicine, Durham, NC, USA. [8]Duke Human Vaccine Institute, Duke University School of Medicine, Durham, NC, USA. [9]New York University School of Medicine, NYU Langone Health, New York, NY, USA. [10]Office of Collaborative Biostatistics, Center for Cancer Research, National Cancer Institute, Bethesda, MD, USA. [11]Alchem Laboratories Corporation, Alachua, FL, USA. [12]Department of Medicine, Division of Infectious Diseases, Icahn School of Medicine at Mount Sinai, NY New York, USA. [13]These authors contributed equally: Mohammad Arif Rahman, Massimiliano Bissa, Hanna Scinto. [14]These authors jointly supervised this work: Genoveffa Franchini, Jay A. Berzofsky. ✉e-mail: franchig@mail.nih.gov; berzofsj@mail.nih.gov

