## [Peer Review file · Nature Communications]

Loss of HIV candidate vaccine efficacy in male macaques by mucosal nanoparticle immunization rescued by V2-specific response

Corresponding Author: Dr Genoveffa Franchini

Version 0:

Reviewer comments:

Reviewer #1

(Remarks to the Author)

The authors describe new results using a novel mucosal vaccine encoding a protective V2 epitope shown in previous studies by the investigators to elicit protective immune responses to repeat, low-dose challenges with SIVmac251. The mucosal vaccine is a novel fusion of the V2-loop epitope with pentameric typhoid toxin (V2-TTB) formulated with a nanoparticle designed to selectively release the vaccine in the intestine. This is a novel aspect of the study that selectively evaluates the ability of the V2-loop TTB scaffold to elicit protective immunity against SIVmac251. The data show that the V2-loop specific immune responses play a critical role in vaccine efficacy by immunological mechanisms other than neutralizing antibodies.

The key data supporting the hypothesis are as follows:

1. Vaccine Efficacy: The study conducted intrarectal SIVmac251 challenges on macaques that were immunized with the $\Delta V1DNA/ALVAC/\Delta V1gp120$ /alum vaccine alone or in combination with V2-TTB NP, TTB NP, or empty NP. The V2-TTB NP group showed a strong trend towards decreased risk of virus acquisition compared to the control groups. This suggests that the presence of V2 in the nanoparticles enhanced vaccine efficacy.
2. Antibody Responses: The V2-TTB NP group exhibited higher levels of V2-specific ADCC antibodies compared to the other groups. ADCC activity and V2-specific ADCC correlated with a decreased risk of virus acquisition. The V2-TTB NP group also had higher levels of envelope-specific antibodies in plasma and mucosa, indicating an enhanced immune response.
3. Mucosal Immune Responses: Mucosal immunization with V2-TTB NP increased the frequency of protective CD14+ cells and NKp44+ ILCs in the rectal mucosa. These cell populations have been associated with decreased risk of infection. In contrast, TTB NP and empty NP immunization increased non-protective NKp44-NKG2A-IFN- γ + cells and activated Ki67+CD4+ T cells, which correlate with increased susceptibility to infection.
4. Impact on Dendritic Cells: Mucosal immunization with TTB NP and empty NP preferentially recruited mucosal plasmacytoid dendritic cells (pDCs), which were associated with faster virus acquisition. In contrast, the V2-TTB NP group showed a reduced frequency of pDCs, suggesting that the presence of V2 mitigated pDC engagement.
5. Impact on NK/ILCs: Mucosal immunization with TTB NP and empty NP decreased the frequency of protective NKp44+ cells, while increasing the frequency of non-protective NKp44-NKG2A-IFN- γ + cells. In contrast, the V2-TTB NP group maintained a higher frequency of protective NKp44+ cells.
6. Impact on CD4+ T Cells: Mucosal immunization with TTB NP and empty NP increased the frequency of activated Ki67+CD4+ T cells in the rectal mucosa, which could serve as targets for virus entry. In contrast, the V2-TTB NP group showed lower levels of activated CD4+ T cells.

These data collectively support the hypothesis that the presence of V2 in the nanoparticles enhances vaccine efficacy by promoting protective immune responses and counteracting the negative effects of nanoparticles.

There are several aspects of the study that deserve additional comment.

1. The document does not explicitly state the virus acquisition rate for the Vaccine+V2-TTB NP group. However, it mentions that the V2-TTB NP group showed a strong trend towards decreased risk of virus acquisition compared to the control groups ($p=0.053$). This suggests that the Vaccine+V2-TTB NP group had a lower virus acquisition rate compared to the control groups, although it did not reach statistical significance in this study.
2. There is extensive use of historical controls due to the complex vaccine regimen. The potential impact of using historical controls on the significance of the findings should be indicated.

3. The vaccine regimens elicit different levels of post-infection control of viremia. This is an important observation that indicates an additional aspect of immunization with V2-TTB with significant implications for vaccine-elicited control of viremia, which deserves additional comment.

Reviewer #2

(Remarks to the Author)

General comment

The ongoing challenge of developing an effective HIV-1 vaccine is critical due to the virus's genetic diversity and the complexity of inducing broadly neutralizing antibodies. Only the RV144 trial has shown modest efficacy, correlating with specific antibody and T-cell responses.

Recent research has focused on the macaque model, where modifications to the Canarypox vaccine platform have shown improved efficacy. Notably, removing V1 epitopes from the SIV envelope immunogens resulted in significant protection against SIV in macaques. This approach, utilizing a DNA/ALVAC/envelope protein/alum platform, decreased the risk of SIV acquisition, emphasizing the importance of mucosal antibodies to V2 and specific immune cell responses in providing protection.

In this study, Arif Rahman et al. hypothesized that systemic immunization could be enhanced by targeting the gut mucosal immune system via oral delivery of a V2 epitope. Using PLGA nanoparticles, the V2 epitope was delivered effectively to the large intestine and rectal mucosa.

The methodology involved several key steps:

1. Vaccine Formulation: Creating the V2 epitope scaffolded with typhoid toxin B subunit (TTB), encapsulated in PLGA nanoparticles.
2. Nanoparticle Coating: Coating nanoparticles with Eudragit FS30D polymer to ensure targeted release in the large intestine.
3. Targeted Delivery: Ensuring that nanoparticles reach the large intestine for uptake and immune response induction. These methods aimed to induce immunity in the colorectal mucosa, as previously demonstrated in mice and macaques. The study assessed the impact on immune cell recruitment and the overall immune response in the mucosal and systemic compartments.

Arif Rahman et al. observed that nanoparticles lacking the V2 epitope negatively impacted vaccine efficacy by recruiting detrimental immune cells and increasing susceptibility to infection. Conversely, animals immunized with V2-TTB nanoparticles showed a strong immune response to V2, counteracting the negative effects and trending toward significant protection.

Arif Rahman et al study could represent a significant advancement in HIV vaccine research, demonstrating the potential of targeted mucosal immunization to enhance vaccine efficacy. By innovatively utilizing PLGA nanoparticles to deliver the V2 epitope directly to the gut mucosal immune system, the researchers may pave the way for more effective vaccination strategies.

The topic of this article is relevant and highlight the importance of such cutting-edge approaches in the ongoing fight against HIV.

Nevertheless The manuscript's structure and the presentation of data are not optimal, leading to difficulties in following the experimental flow and interpreting the results clearly. moreover the manuscript suffers from a significant lack of methodological detail, with major experimental weaknesses that need to be addressed. Additionally, some conclusions should be moderated to avoid misinterpretation of the data.

Major concern

In figure 2

The term 'ADCC titer' is not commonly used in scientific literature and is confusing. To what does the term 'ADCC titer' refer? Are the authors referring to the concentration or level of all antibodies in a sample (such as blood serum) capable of inducing ADCC, or are they specifically discussing V2-specific antibodies that can perform ADCC? If it corresponds to a measure of antibody concentration, why do the authors not use the term 'antibody concentration'? Additionally, the term 'measurement of ADCC titer' is not mentioned in the Methods section, and reference 28 does not detail the method for measuring ADCC titer. This needs clarification.

Could the authors please provide a correlation analysis using the ADCC titer measured across all monkeys, irrespective of their vaccination groups?

Did the authors take into consideration the IgG concentration in the plasma of all monkeys to standardize their assays across animals? This aspect is not mentioned in the Methods section. The same observation applies to the V2-specific ADCC activity

In the context of V2-specific ADCC activity, it would be valuable to know the proportion of V2-specific antibodies relative to total IgG in the plasma of different monkeys. This information could help assess the impact of purified F(ab')₂ fragments from NCI05 in the blockade assay.

In the context of V2-specific ADCC activity, what percentage of killing was measured between conditions?

The concept of specific ADCC activity can be confusing. If the authors intend to use this term, they should clarify that the assays were performed using purified V2-specific antibodies?

Did the authors investigate the possibility of testing different concentrations of purified F(ab')₂ fragments from the NCI05 monoclonal antibody? If so, the corresponding data should be included

The ADCC activity, both with and without F(ab')₂ fragments from NCI05, should be presented (e.g., in the supplementary Figures). This would greatly aid in understanding the assay.

In Figure 2E, the correlation does not appear appropriate. To fully assess the impact of ADCC activity on the number of rectal challenges needed to infect the monkeys, correlations should be conducted using values obtained from all monkeys, regardless of their vaccination status.

It is expected that following vaccination, the avidity of antibodies typically increases over time due to affinity maturation. In this context, the differences observed between groups in Figure 2F are challenging to attribute solely to the vaccination strategy. Could the authors provide the comparison at the same time point?

In the context of HIV's low envelope spike density as a sophisticated evolutionary adaptation to evade the immune response by reducing the effectiveness of bivalent antibody binding, could the authors discuss the relevance of measuring antibody avidity in this figure?

The authors appear to suggest a link between high avidity and high ADCC potential. However, a previous study (doi: 10.1128/JVI.02544-12) demonstrated that although vaccine-induced protection from SIVmac251 acquisition was significantly associated with antibodies having high avidity for gp120 and neutralizing activity in vitro, there was no significant correlation with ADCC or T-cell responses measured by ELISpot, intracellular cytokine staining, and T-cell proliferation. Could the authors discuss this discrepancy?

Did the authors perform flow analysis of PBMC composition before conducting ADCC activity assays? This information would be valuable for interpreting the results, considering that not all cell types are capable of ADCC.

Was the ADCC activity performed on fresh or frozen cells? Considering that the systemic vaccine group was studied previously, this point is not clarified in the Methods section and could impact the interpretation of the results.

In Figure 3B, what does trogocytosis correspond to? Does it refer to the percentage of CD14+ cells positive for PKH26? Was the assay performed with an antibody-free control?

In this assay, how do the authors discriminate between cells that have undergone trogocytosis and cells that have performed ADCP? Could the authors clarify this distinction? Additionally, the use of confocal microscopy would be valuable for evaluating trogocytosis activity

Performing flow cytometry analysis on rectal biopsy samples presents several challenges that can impact the quality and reliability of the data obtained. In this context, the authors should clarify certain aspects of their methodology:

- Line 1213: How many rectal biopsies were performed per time point?
- Line 1216: How many total cells were recovered following mechanical and chemical processing? Low cell yield can affect the accuracy and reliability of flow cytometry results.
- Line 1217: The authors mentioned that a portion of the cells were phenotyped; could they clarify this point? Some cell surface markers are sensitive to enzymatic digestion and may be cleaved or altered, potentially leading to inaccurate detection or loss of important markers. Has this aspect been considered?
- Line 1217: The authors should provide the entire gating strategy, including dot plots with the numbers of cells in each quadrant instead of smooth contour plots and percentages.

The generally accepted minimum number of events (cells) needed to perform meaningful analysis and generate statistically robust data is at least 300 events per cell population of interest. Therefore, to facilitate the interpretation of the data, the authors should provide the absolute count of each studied population in the manuscript and exclude from the analysis any population with a count of fewer than 300 cells.

Was flow analysis performed on fresh or frozen cells? Considering that the systemic vaccine group was studied in a previous study, this point is not clarified in the Methods section and could impact the interpretation of the results.

Line 323: The authors performed immunohistochemistry staining of Ki67+CD4+ T cells in rectal mucosa, as presented in Figure 4K. It would be valuable for the authors to include images of the staining performed in the manuscript. This could help support the interpretation made in Figure 4K. Additionally, could the authors provide a clear description of the methods used to measure the number of Ki67+CD4+ T cells in rectal mucosa, including details such as the number of biopsies used, the number of animals, the number of images counted, etc.?

Minor concern

Line 88, To facilitate reading, could the authors clarify which cell subsets the NKP44-NKG2a- cells refer to (e.g., ILC, NK cells, or others)?

Line 113: Could the authors clarify which cell populations they are referring to when mentioning NKP44+ cells (e.g., ILC)?

Line 118: Could the authors specify and list what is meant by the increased mucosal and systemic immune response?

Figure 1 To facilitate the interpretation of the results, the comparison between the systemic vaccination group and the untreated, combined controls group should be discussed first in the Results section. Additionally, these two groups should be presented first in Panel A, and Panels B and C should be interchanged.

In Figure 1, Panels D to F should be rearranged so that the letters indicating the panels appear before the legends of the panels.

In Panel D, the comparison refers to "vaccine + V2 NP." Do the authors mean "vaccine + V2_TTB NP" as mentioned in line 152?

In figure 2, Overall, the data presented in this figure should be reorganized to facilitate the interpretation of the results. It would be more valuable if the function(ADCC part) is presented after the antibody titration.

in Line 172, avidity is mentioned, but in the related figure, it is referred to as avidity score. Could the authors standardize the terminology?

The data from Supplementary Figure 2d should be incorporated into Figure 2.

In the chapter on mucosal CD14+ cells rescued by oral V2-TTB NP immunization, the authors reference Supplementary Fig. 4a (line 202) before Supplementary Fig. 3a (line 223)

Due to my expertise limitations, I am unable to evaluate the assay regarding the characterization of CD14+ efferocytes.

Therefore, this part of the manuscript was not reviewed.

In line 207, the authors claimed that trogocytosis is an antibody-dependent process. However, trogocytosis is primarily understood as a process involving the transfer of membrane fragments or molecules from one cell to another. While it can occur in various contexts, including interactions between immune cells and target cells, its dependence on antibodies is not universal. Therefore, could the authors revise this sentence?

Lines 213-214: Do the authors refer to in vivo or in vitro conditions? This sentence must be documented or deleted, as it appears speculative.

Version 1:

Reviewer comments:

Reviewer #1

(Remarks to the Author)

The authors have responded appropriately to my comments. No further comments.

Reviewer #2

(Remarks to the Author)

I appreciate the substantial effort you have invested in addressing the majority of the comments and suggestions raised during the review process. Most of the concerns I previously mentioned have been thoroughly addressed, and I commend you on the improvements made to the manuscript. However, there are still a few points that require further clarification or elaboration to enhance the overall quality and scientific rigor of the paper.

The authors stated that ADCC was consistently performed using PBMCs from the same human donor as effector cells. While this approach ensures consistency, PBMCs from a single donor may not capture the full spectrum of immune responses observed across a broader population. Immune cell functionality can vary significantly between individuals due to factors such as genetics, environment, and health status. As a result, the findings may not be generalizable to other populations or donor samples. Additionally, the donor's specific immune profile may influence the results, and different donors could exhibit varying ADCC responses, potentially leading to different conclusions.

A more robust approach might involve incorporating PBMCs from multiple donors to account for inter-individual variability, while still maintaining a controlled experimental design. This would increase the representativeness and generalizability of the findings.

Furthermore, the authors mentioned that the trogocytosis assay was performed with an antibody-free control. It would be beneficial to present these results and directly compare them to the condition including the antibody to strengthen the interpretation of the data.

Lastly, could the authors please verify the accuracy of reference Navarro et al., 2024 (Ref. 86)? It appears that this reference does not pertain to the isolation of tissue cells via mechanical or enzymatic processes, as suggested.

Thank you for the opportunity to revise this manuscript. We wish to thank the reviewers for their careful evaluation of the work presented.
Please see our point-by-point response below.

Reviewer #1 (Remarks to the Author):

The authors describe new results using a novel mucosal vaccine encoding a protective V2 epitope shown in previous studies by the investigators to elicit protective immune responses to repeat, low-dose challenges with SIVmac251. The mucosal vaccine is a novel fusion of the V2-loop epitope with pentameric typhoid toxin (V2-TTB) formulated with a nanoparticle designed to selectively release the vaccine in the intestine. This is a novel aspect of the study that selectively evaluates the ability of the V2-loop TTB scaffold to elicit protective immunity against SIVmac251. The data show that the V2-loop specific immune responses play a critical role in vaccine efficacy by immunological mechanisms other than neutralizing antibodies. The key data supporting the hypothesis are as follows:

1. Vaccine Efficacy: The study conducted intrarectal SIVmac251 challenges on macaques that were immunized with the Δ V1DNA/ALVAC/ Δ V1gp120/alum vaccine alone or in combination with V2-TTB NP, TTB NP, or empty NP. The V2-TTB NP group showed a strong trend towards decreased risk of virus acquisition compared to the control groups. This suggests that the presence of V2 in the nanoparticles enhanced vaccine efficacy.
2. Antibody Responses: The V2-TTB NP group exhibited higher levels of V2-specific ADCC antibodies compared to the other groups. ADCC activity and V2-specific ADCC correlated with a decreased risk of virus acquisition. The V2-TTB NP group also had higher levels of envelope-specific antibodies in plasma and mucosa, indicating an enhanced immune response.
3. Mucosal Immune Responses: Mucosal immunization with V2-TTB NP increased the frequency of protective CD14⁺ cells and NKp44⁺ ILCs in the rectal mucosa. These cell populations have been associated with decreased risk of infection. In contrast, TTB NP and empty NP immunization increased non-protective NKp44-NKG2A-IFN- γ ⁺ cells and activated Ki67⁺CD4⁺ T cells, which correlate with increased susceptibility to infection.
4. Impact on Dendritic Cells: Mucosal immunization with TTB NP and empty NP preferentially recruited mucosal plasmacytoid dendritic cells (pDCs), which were associated with faster virus acquisition. In contrast, the V2-TTB NP group showed a reduced frequency of pDCs, suggesting that the presence of V2 mitigated pDC engagement.
5. Impact on NK/ILCs: Mucosal immunization with TTB NP and empty NP decreased the frequency of protective NKp44⁺ cells, while increasing the frequency of non-protective NKp44-NKG2A-IFN- γ ⁺ cells. In contrast, the V2-TTB NP group maintained a higher frequency of protective NKp44⁺ cells.
6. Impact on CD4⁺ T Cells: Mucosal immunization with TTB NP and empty NP increased the frequency of activated Ki67⁺CD4⁺ T cells in the rectal mucosa, which could serve as targets for virus entry. In contrast, the V2-TTB NP group showed lower levels of activated CD4⁺ T cells. These data collectively support the hypothesis that the presence of V2 in the nanoparticles enhances vaccine efficacy by promoting protective immune responses and counteracting the negative effects of nanoparticles.

There are several aspects of the study that deserve additional comment.

1. The document does not explicitly state the virus acquisition rate for the Vaccine+V2-TTB NP group. However, it mentions that the V2-TTB NP group showed a strong trend towards decreased risk of virus acquisition compared to the control groups ($p=0.053$). This suggests that the Vaccine+V2-TTB NP group had a lower virus acquisition rate compared to the control groups, although it did not reach statistical significance in this study.

We added the difference in the average risk of SIV_{mac251} acquisition (as a readout of vaccine efficacy) of the V2-TTB-NP group (47.6%) versus the naïve controls as requested. (Abstract and Line 155).

2. There is extensive use of historical controls due to the complex vaccine regimen. The potential impact of using historical controls on the significance of the findings should be indicated.

As we state in line 150, a small number of concurrent controls were used to validate virus infectivity, but the plan *a priori* was to combine the concurrent and historical controls if we observed no difference. As demonstrated in Supplementary Fig. 1A, the acquisition curves were almost identical, thus warranting the combination of data.

3. The vaccine regimens elicit different levels of post-infection control of viremia. This is an important observation that indicates an additional aspect of immunization with V2-TTB with significant implications for vaccine-elicited control of viremia, which deserves additional comment.

We observed V2-specific ADCC activity was associated with VL in the systemic vaccine group (**Fig. 2k**), whereas no such correlation was observed for NP treated groups. Trogocytosis was positively associated with VL in the V2-TTB NP group (**Fig. 3c**), and negatively associated with VL in the systemic vaccine group (**Fig. 3d**), suggesting these antibody-dependent monocyte responses act differently in viremia control in different vaccine regimens. Furthermore, mDCs were associated with VL control in the V2-TTB NP group (**Fig. 3e**), whereas pDCs promoted increased VL in TTB NP (**Fig. 3h**). Interestingly, PMA/Ionomycin-induced IFN- γ^+ NKG2A⁻ NKp44⁻ ILCs promoted VL increase in V2-TTB and TTB group (**Fig. 3l, m**). Taken together, these data suggested that different vaccine regimens promote distinct immune responses, which either are associated with viremia control or the promotion of VL. (**Lines 524-534**).

Reviewer #2 (Remarks to the Author):

General comment

The ongoing challenge of developing an effective HIV-1 vaccine is critical due to the virus's genetic diversity and the complexity of inducing broadly neutralizing antibodies. Only the RV144 trial has shown modest efficacy, correlating with specific antibody and T-cell responses. Recent research has focused on the macaque model, where modifications to the Canarypox vaccine platform have shown improved efficacy. Notably, removing V1 epitopes from the SIV envelope immunogens resulted in significant protection against SIV in macaques. This approach,

utilizing a DNA/ALVAC/envelope protein/alum platform, decreased the risk of SIV acquisition, emphasizing the importance of mucosal antibodies to V2 and specific immune cell responses in providing protection.

In this study, Arif Rahman et al. hypothesized that systemic immunization could be enhanced by targeting the gut mucosal immune system via oral delivery of a V2 epitope. Using PLGA nanoparticles, the V2 epitope was delivered effectively to the large intestine and rectal mucosa. The methodology involved several key steps:

1. Vaccine Formulation: Creating the V2 epitope scaffolded with typhoid toxin B subunit (TTB), encapsulated in PLGA nanoparticles.
2. Nanoparticle Coating: Coating nanoparticles with Eudragit FS30D polymer to ensure targeted release in the large intestine.
3. Targeted Delivery: Ensuring that nanoparticles reach the large intestine for uptake and immune response induction.

These methods aimed to induce immunity in the colorectal mucosa, as previously demonstrated in mice and macaques. The study assessed the impact on immune cell recruitment and the overall immune response in the mucosal and systemic compartments.

Arif Rahman et al. observed that nanoparticles lacking the V2 epitope negatively impacted vaccine efficacy by recruiting detrimental immune cells and increasing susceptibility to infection. Conversely, animals immunized with V2-TTB nanoparticles showed a strong immune response to V2, counteracting the negative effects and trending toward significant protection. Arif Rahman et al study could represent a significant advancement in HIV vaccine research, demonstrating the potential of targeted mucosal immunization to enhance vaccine efficacy. By innovatively utilizing PLGA nanoparticles to deliver the V2 epitope directly to the gut mucosal immune system, the researchers may pave the way for more effective vaccination strategies. The topic of this article is relevant and highlight the importance of such cutting-edge approaches in the ongoing fight against HIV.

Nevertheless The manuscript's structure and the presentation of data are not optimal, leading to difficulties in following the experimental flow and interpreting the results clearly. moreover the manuscript suffers from a significant lack of methodological detail, with major experimental weaknesses that need to be addressed. Additionally, some conclusions should be moderated to avoid misinterpretation of the data.

Major concern

In figure 2

1. The term 'ADCC titer' is not commonly used in scientific literature and is confusing. To what does the term 'ADCC titer' refer? Are the authors referring to the concentration or level of all antibodies in a sample (such as blood serum) capable of inducing ADCC, or are they specifically discussing V2-specific antibodies that can perform ADCC? If it corresponds to a measure of antibody concentration, why do the authors not use the term 'antibody concentration'? Additionally, the term 'measurement of ADCC titer' is not mentioned in the Methods section, and reference 28 does not detail the method for measuring ADCC titer. This needs clarification.

ADCC titer is measured by serial dilution of plasma. ADCC titer refers to the lowest dilution at which ADCC killing can be detected in plasma. The ADCC endpoint titer is defined as the

reciprocal dilution at which the percentage of ADCC killing was greater than the mean percentage of killing of the negative control wells containing medium, target cells, and effector cells, plus three standard deviations. The methods section “ADCC CEM-based assay” describes ADCC titer (L1157). We have added an additional reference for the assay (ref. 77, Gomez-Roman *et al.* 2006). Also see references 78-79, Helmold-Hait *et al.* 2020, Musich *et al.* 2020.

The ADCC titer is now defined in the Results section line 185 where Fig 2d is described, as well as in the Figure 2d legend.

2. Could the authors please provide a correlation analysis using the ADCC titer measured across all monkeys, irrespective of their vaccination groups?

We have provided the correlation analysis for ADCC titer in Supplementary Figure 2d as requested. The correlation with number of challenges required to infect was significant at the $p=0.0001$ level with $R = 0.59$.

3. Did the authors take into consideration the IgG concentration in the plasma of all monkeys to standardize their assays across animals? This aspect is not mentioned in the Methods section.

We have used monoclonal antibodies as positive control and medium as negative control across assays and for normalization of the assay. In the Methods section, lines 1139-42 indicate “IgG purified from a SIVmac251-infected rhesus macaque was included as an assay positive control and blank (uncoupled) and empty gp70 scaffold (MuLV gp70) conjugated beads as negative controls.” For serum antibodies, levels in individual animals were not normalized to that animal’s total IgG, but for mucosal samples, the binding antibodies were normalized to total IgG (Methods lines 1148-51).

4. The same observation applies to the V2-specific ADCC activity. In the context of V2-specific ADCC activity, it would be valuable to know the proportion of V2-specific antibodies relative to total IgG in the plasma of different monkeys. This information could help assess the impact of purified F(ab')₂ fragments from NCI05 in the blockade assay.

We have provided a graph in Supplementary Fig. 2g.

5. In the context of V2-specific ADCC activity, what percentage of killing was measured between conditions?

We have provided a graph in Supplementary Fig. 2f.

6. The concept of specific ADCC activity can be confusing. If the authors intend to use this term, they should clarify that the assays were performed using purified V2-specific antibodies?

We have clarified the nature of ADCC assays (L193-195): “In order to understand the effect of V2-specific ADCC, we next isolated the F(ab')₂ region of a V2 specific monoclonal antibody

and used it to block ADCC activity.” The method is described in the Methods, “Inhibition of ADCC CEM-based assay by monoclonal F(ab')₂ of NCI05,” lines 1181-1194.

7. Did the authors investigate the possibility of testing different concentrations of purified F(ab')₂ fragments from the NCI05 monoclonal antibody? If so, the corresponding data should be included

We have provided a graph in Supplementary Fig. 2e.

8. The ADCC activity, both with and without F(ab')₂ fragments from NCI05, should be presented (e.g., in the supplementary Figures). This would greatly aid in understanding the assay.

We have provided a graph in Supplementary Fig. 2f.

9. In Figure 2E, the correlation does not appear appropriate. To fully assess the impact of ADCC activity on the number of rectal challenges needed to infect the monkeys, correlations should be conducted using values obtained from all monkeys, regardless of their vaccination status.

We have provided a graph in Supplementary Fig. 2d.

10. It is expected that following vaccination, the avidity of antibodies typically increases over time due to affinity maturation. In this context, the differences observed between groups in Figure 2F are challenging to attribute solely to the vaccination strategy. Could the authors provide the comparison at the same time point?

The time course of vaccination was different between systemic and NP groups. The systemic vaccine group had the last boost on week 12, thus week 17 was 5 weeks later, whereas the other groups were boosted again on week 16, so week 24 was only 8 weeks after the last boost. Thus, the difference is not as large as the difference between 17 and 24 weeks, but only 3 weeks more (8-5 weeks). We show in Supplementary Fig. 2h that the portion of the V2-specific responses in gp120-specific antibody titer was comparable among groups except for the empty NP group, suggesting the antibody responses are not affected significantly by avidity maturation after the last boost.

11. In the context of HIV's low envelope spike density as a sophisticated evolutionary adaptation to evade the immune response by reducing the effectiveness of bivalent antibody binding, could the authors discuss the relevance of measuring antibody avidity in this figure?

We thank the reviewer for this suggestion. As requested, we have added some discussion of this point in the Discussion, lines 514-516: “The avidity of the V2 antibody may play a critical role, especially in view of SIV’s low envelope spike density, which reduces the benefit of antibody bivalency and makes ADCC activity more dependent on the intrinsic affinity”

12. The authors appear to suggest a link between high avidity and high ADCC potential. However, a previous study (doi: 10.1128/JVI.02544-12) demonstrated that although vaccine-

induced protection from SIVmac251 acquisition was significantly associated with antibodies having high avidity for gp120 and neutralizing activity in vitro, there was no significant correlation with ADCC or T-cell responses measured by ELISpot, intracellular cytokine staining, and T-cell proliferation. Could the authors discuss this discrepancy?

The vaccine discussed in Pegu 2013 is different from the vaccine we use in the current study. In our earlier study, we used the ALVAC/gp120 vaccine regimen. In the current study, we used Δ V1-DNA/ALVAC/gp120 vaccine regimen. Since the prime and boosting are different, and the latter vaccine regimen does not show V1 specific responses, comparing the data is not ideal.

13. Did the authors perform flow analysis of PBMC composition before conducting ADCC activity assays? This information would be valuable for interpreting the results, considering that not all cell types are capable of ADCC.

ADCC was performed always using same human donor PBMC as effector cells. Thus, the differences observed in the study are not attributed to PBMC.

14. Was the ADCC activity performed on fresh or frozen cells? Considering that the systemic vaccine group was studied previously, this point is not clarified in the Methods section and could impact the interpretation of the results.

ADCC was performed using same human donor PBMC as effector cells and gp120 coated using EGFP-CEM-NKr-CCR5-SNAP cells were used as target cells and frozen plasma as source of antibody. Thus, the differences observed in the study are not attributed to PBMC or target cells, neither of which came from the individual animals. Only the frozen plasma antibody came from individual animals, not any cells in the ADCC assay. This is now clarified in the Methods section lines 1170-1172.

15. In Figure 3B, what does trogocytosis correspond to? Does it refer to the percentage of CD14⁺ cells positive for PKH26? Was the assay performed with an antibody-free control? In this assay, how do the authors discriminate between cells that have undergone trogocytosis and cells that have performed ADCP? Could the authors clarify this distinction? Additionally, the use of confocal microscopy would be valuable for evaluating trogocytosis activity

In Figure 3B, trogocytosis corresponds to percentage of CD14⁺ cells positive for PKH26. We have updated the Y axis of this figure to include this information. The assay was performed with an antibody-free control.

Kramski et al (J Immunol Methods 2012) have used a similar assay and elegantly shown using deconvolution microscopy that monocytes within PBMCs are physically interacting with CEM target cells to take up PKH26 without evidence of phagocytosis. For these reasons, we are confident that our assay is measuring trogocytosis.

16. Performing flow cytometry analysis on rectal biopsy samples presents several challenges that can impact the quality and reliability of the data obtained. In this context, the authors should

clarify certain aspects of their methodology:

- Line 1213: How many rectal biopsies were performed per time point?

We have clarified in the manuscript that 11 rectal pinches were collected per timepoint.

- Line 1216: How many total cells were recovered following mechanical and chemical processing? Low cell yield can affect the accuracy and reliability of flow cytometry results.

We recovered 10-15 million cells from the pinches. Two million cells were used per condition.

- Line 1217: The authors mentioned that a portion of the cells were phenotyped; could they clarify this point? Some cell surface markers are sensitive to enzymatic digestion and may be cleaved or altered, potentially leading to inaccurate detection or loss of important markers. Has this aspect been considered?

Two million cells were used per condition. For flow cytometry analysis, no other method is known to isolate cells from rectal mucosa. We have tried to isolate cells by mechanical process only, absent enzymatic process, but this approach fails to produce high quality single cells. A similar observation was reported by Navarro *et al.* 2024 (Ref. 86). All animals went through the same process of tissue collection and the same methods were used to isolate the cells. Thus, the data are comparable. This is now described in the Methods lines 1313-1320.

- Line 1217: The authors should provide the entire gating strategy, including dot plots with the numbers of cells in each quadrant instead of smooth contour plots and percentages. The generally accepted minimum number of events (cells) needed to perform meaningful analysis and generate statistically robust data is at least 300 events per cell population of interest. Therefore, to facilitate the interpretation of the data, the authors should provide the absolute count of each studied population in the manuscript and exclude from the analysis any population with a count of fewer than 300 cells.

We have updated the gating strategy in dot plots (Supplementary Fig 3). We have also excluded the data which had lower than 300 events per cell population of interest. Thus, CD14⁺, CD16⁺ and CD14⁺CD16⁺ monocytes and gp120-specific IL-17⁺NKp44 cells were excluded, since in most cases the cell population of interest had lower than 300 events. We have provided the absolute count of each reported rectal population in the manuscript in the supplementary table.

Was flow analysis performed on fresh or frozen cells? Considering that the systemic vaccine group was studied in a previous study, this point is not clarified in the Methods section and could impact the interpretation of the results.

The rectal mucosal flowcytometry analysis was performed on fresh cells. We did not include any mucosal cell data from systemic vaccine group, which was studied in a previous study. We don't have data from the historical systemic vaccine group.

Line 323: The authors performed immunohistochemistry staining of Ki67+CD4+ T cells in rectal mucosa, as presented in Figure 4K. It would be valuable for the authors to include images of the

staining performed in the manuscript. This could help support the interpretation made in Figure 4K.

We have incorporated the images as Figure 4L.

Additionally, could the authors provide a clear description of the methods used to measure the number of Ki67+CD4+ T cells in rectal mucosa, including details such as the number of biopsies used, the number of animals, the number of images counted, etc.?

Multiple layer images with CD4, Ki67, and DAPI stains in separated layers were captured on 5 randomly selected fields covering 472100 μm^2 of tissue. One transparency reference layer was generated to mark CD4, Ki67, and CD4/Ki67 double-positive cells with DAPI nuclear stain. Also, a merged image of CD4, Ki67, and DAPI stain was used to eliminate unspecific stains. The following number of animals were used for the respective group, V2-TTB NP (n=12), TTB NP (n=6), empty NP (n=9) and systemic vaccine (n=14).

Minor concern

Line 88, To facilitate reading, could the authors clarify which cell subsets the NKP44-NKG2a-cells refer to (e.g., ILC, NK cells, or others)?

We have clarified these as ILCs in the whole manuscript

Line 113: Could the authors clarify which cell populations they are referring to when mentioning NKP44+ cells (e.g., ILC)?

We have clarified these as ILCs in the whole manuscript

Line 118: Could the authors specify and list what is meant by the increased mucosal and systemic immune response?

We have added this information at lines 121-123.

Figure 1 To facilitate the interpretation of the results, the comparison between the systemic vaccination group and the untreated, combined controls group should be discussed first in the Results section. Additionally, these two groups should be presented first in Panel A, and Panels B and C should be interchanged.

We have made the suggested changes.

In Figure 1, Panels D to F should be rearranged so that the letters indicating the panels appear before the legends of the panels.

We have made the suggested changes.

In Panel D, the comparison refers to "vaccine + V2 NP." Do the authors mean "vaccine + V2_TTB NP" as mentioned in line 152?

Yes; we have made the correction.

In figure 2, Overall, the data presented in this figure should be reorganized to facilitate the interpretation of the results. It would be more valuable if the function (ADCC part) is presented after the antibody titration.

We have made the suggested changes.

In Line 172, avidity is mentioned, but in the related figure, it is referred to as avidity score. Could the authors standardize the terminology?

We have made the changes and clarified the meaning of avidity score.

The data from Supplementary Figure 2d should be incorporated into Figure 2.

We have made the suggested changes.

In the chapter on mucosal CD14+ cells rescued by oral V2-TTB NP immunization, the authors reference Supplementary Fig. 4a (line 202) before Supplementary Fig. 3a (line 223)

We have updated the figure numbers.

Due to my expertise limitations, I am unable to evaluate the assay regarding the characterization of CD14+ efferocytes. Therefore, this part of the manuscript was not reviewed.

We appreciate the reviewer for it.

In line 207, the authors claimed that trogocytosis is an antibody-dependent process. However, trogocytosis is primarily understood as a process involving the transfer of membrane fragments or molecules from one cell to another. While it can occur in various contexts, including interactions between immune cells and target cells, its dependence on antibodies is not universal. Therefore, could the authors revise this sentence?

We have removed "antibody-dependent process" from the sentence.

Lines 213-214: Do the authors refer to in vivo or in vitro conditions? This sentence must be documented or deleted, as it appears speculative.

We removed the sentence.

Thank you for the opportunity to revise this manuscript. We wish to thank the reviewers for their careful evaluation of the work presented. Please see our point-by-point response below.

Reviewer #1 (Remarks to the Author):

The authors have responded appropriately to my comments. No further comments.

We thank the reviewer for the evaluation of the manuscript.

Reviewer #2 (Remarks to the Author):

I appreciate the substantial effort you have invested in addressing the majority of the comments and suggestions raised during the review process. Most of the concerns I previously mentioned have been thoroughly addressed, and I commend you on the improvements made to the manuscript. However, there are still a few points that require further clarification or elaboration to enhance the overall quality and scientific rigor of the paper.

The authors stated that ADCC was consistently performed using PBMCs from the same human donor as effector cells. While this approach ensures consistency, PBMCs from a single donor may not capture the full spectrum of immune responses observed across a broader population. Immune cell functionality can vary significantly between individuals due to factors such as genetics, environment, and health status. As a result, the findings may not be generalizable to other populations or donor samples. Additionally, the donor's specific immune profile may influence the results, and different donors could exhibit varying ADCC responses, potentially leading to different conclusions. A more robust approach might involve incorporating PBMCs from multiple donors to account for inter-individual variability, while still maintaining a controlled experimental design. This would increase the representativeness and generalizability of the findings.

We agree with the reviewer that using PBMCs from different donors may affect the frequency of ADCC killing. Due to donor-to-donor variability, the ADCC killing rate would proportionally adjust based on the specific PBMC used. Thus, data from different donors should be normalized for accurate comparison. To account for this variability, testing multiple donors with a sufficient number of plasma samples is necessary. However, using PBMCs from multiple donors is aimed at reducing inter-experiment variability, which can also be controlled by using a single donor with a larger sample size. In this study, we used PBMCs from a single donor for all ADCC assays, ensuring that the data presented are consistent and comparable.

Furthermore, the authors mentioned that the trogocytosis assay was performed with an antibody-free control. It would be beneficial to present these results and directly compare them to the condition including the antibody to strengthen the interpretation of the data.

Figure 3b has been normalized based on Antibody-free control. The data of Antibody-free control is shared in the sourced data file.

Lastly, could the authors please verify the accuracy of reference Navarro et al., 2024 (Ref. 86)? It appears that this reference does not pertain to the isolation of tissue cells via mechanical or enzymatic processes, as suggested.**

We apologize for the mistake. We have corrected the reference:

*Bondonese, A., et al. Impact of enzymatic digestion on single cell suspension yield from peripheral human lung tissue. Cytometry A **103**, 777-785 (2023).*